# OmniCT: Towards a Unified Slice-Volume LVLM for Comprehensive CT Analysis

**Tianwei Lin**[1,2*], **Zhongwei Qiu**[2,3,1*], **Wenqiao Zhang**[1†], **Jiang Liu**[1], **Yihan Xie**[1],
**Mingjian Gao**[1], **Zhenxuan Fan**[1], **Zhaocheng Li**[1], **Sijing Li**[1,2], **Zhongle Xie**[1],
**Peng Lu**[1], **Yueting Zhuang**[1], **Ling Zhang**[2], **Beng Chin Ooi**[1], **Yingda Xia**[2]
[1]Zhejiang University, [2]DAMO Academy, Alibaba Group, [3]Hupan Lab
{lintw, wenqiaozhang}@zju.edu.cn
{qiuzhongwei.qzw}@alibaba-inc.com

## Abstract

Computed Tomography (CT) is one of the most widely used and diagnostically information-dense imaging modalities, covering critical organs such as the heart, lungs, liver, and colon. Clinical interpretation relies on both *slice-driven* local features (e.g., sub-centimeter nodules, lesion boundaries) and *volume-driven* spatial representations (e.g., tumor infiltration, inter-organ anatomical relations). However, existing Large Vision–Language Models (LVLMs) remain fragmented in CT slice versus volumetric understanding: slice-driven LVLMs show strong generalization but lack cross-slice spatial consistency, while volume-driven LVLMs explicitly capture volumetric semantics but suffer from coarse granularity and poor compatibility with slice inputs. The absence of a unified modeling paradigm constitutes a major bottleneck for the clinical translation of medical LVLMs. We present **OmniCT**, a powerful unified slice–volume LVLM for CT scenarios, which makes three contributions: **(i) Spatial Consistency Enhancement (SCE):** volumetric slice composition combined with tri-axial positional embedding that introduces volumetric consistency, and an MoE hybrid projection enables efficient slice–volume adaptation; **(ii) Organ-level Semantic Enhancement (OSE):** segmentation and ROI localization explicitly align anatomical regions, emphasizing lesion- and organ-level semantics; **(iii) MedEval-CT:** the largest slice–volume CT dataset and hybrid benchmark integrates comprehensive metrics for unified evaluation. OmniCT consistently outperforms existing methods with a substantial margin across diverse clinical tasks and satisfies both micro-level detail sensitivity and macro-level spatial reasoning. More importantly, it establishes a new paradigm for cross-modal medical imaging understanding. Our project is available at https://github.com/ZJU4HealthCare/OmniCT.

## 1 Introduction

Large Vision–Language Models (LVLMs) have become a cornerstone of multi-modal artificial intelligence, demonstrating strong cross-modal representation and reasoning capabilities in both image understanding (Qiu et al., 2023; Li et al., 2024a; Zhu et al., 2025; Bai et al., 2025b) and video perception (Lin et al., 2023a; Li et al., 2024a; Zhang et al., 2025a; Yuan et al., 2025). Benefiting from large-scale pre-training and modality alignment, LVLMs achieve remarkable performance in open-domain tasks (Qiu et al., 2022; Yue et al., 2024; Fu et al., 2025), excelling in both generation and reasoning. These advances establish LVLMs as a universal paradigm for unified vision–language modeling, where the joint modeling of 2D and 3D modalities has emerged as a key design principle.

In recent years, the potential of LVLMs in medical imaging has received increasing attention, with exploration in radiological imaging being particularly notable (Wu et al., 2025; Xu et al., 2025). However, most existing methods are tailored to process either CT slices (Chen et al., 2024a; Lin et al., 2025) or volumetric data (Bai et al., 2024; Hamamci et al., 2024c), with limited focus on

---

*Equal contributions. The work was done during Tianwei's internship at DAMO Academy.
†Corresponding author.

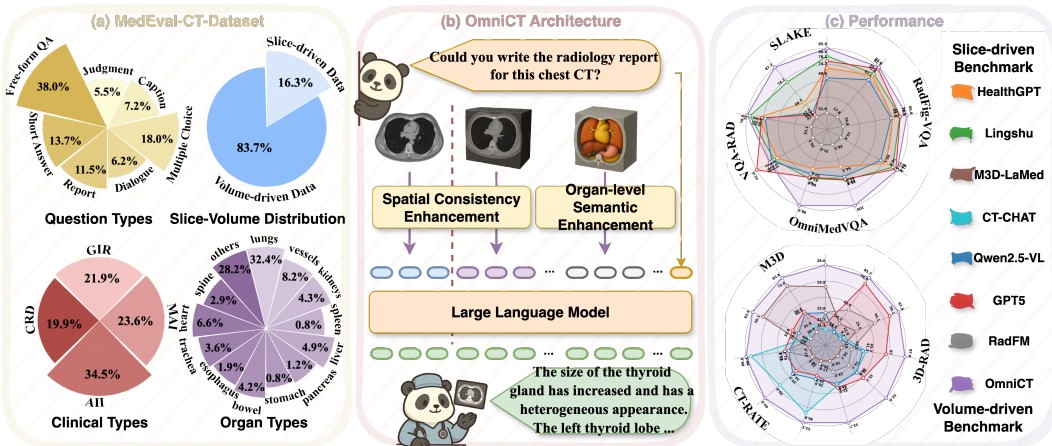

Figure 1: (a) is the statistics of the proposed MedEval-CT-Dataset. (b) describes the simplified architecture of proposed OmniCT. (c) shows that OmniCT consistently surpasses all baselines on both slice-driven and volume-driven CT benchmarks.

cooperative processing. Slice-driven models leverage large-scale 2D pre-training to achieve strong vision–language alignment and perform well in tasks such as lesion detection and radiology report description, yet they fail to capture cross-slice spatial consistency. In contrast, volume-driven models explicitly model voxel-level spatial structures, offering advantages in holistic spatial representation and organ-level reasoning. Nevertheless, these models often lack sensitivity to fine-grained abnormalities and boundary morphology, and their architectures are difficult to adapt to slice-level tasks, thereby limiting their applicability across diverse medical scenarios. This persistent dichotomy between slice and volume modeling constitutes a major bottleneck in the development of medical LVLMs.

Among various medical imaging modalities, CT is one of the most widely used and dense in information, with hundreds of million performed each year globally. CT can cover critical organs such as the heart, lungs, liver, and colon, and is widely applied in essential tasks including disease screening (Hu et al., 2025), lesion assessment (Li et al., 2025b; Shui et al., 2025), and tumor staging (Bassi et al., 2025; Qiu et al., 2025). Its diagnostic process relies on both slice-level local imaging cues, such as sub-centimeter pulmonary nodules or hepatic lesion boundaries, and volume-level spatial–topological representations, such as tumor infiltration ranges or inter-organ anatomical relationships. Modeling along a single dimension alone cannot meet these dual requirements. Therefore, integrating the complementary strengths of 2D and 3D modeling within a unified framework is not only a central scientific challenge for CT understanding but also an inevitable step toward the clinical translation of medical LVLMs.

We propose **OmniCT** (see Fig. 1), a powerful unified slice–volume LVLM for CT-centric understanding, which preserves the cross-modal alignment and generalization strengths of 2D models while integrating the spatial structural awareness of 3D models. To bridge the modality gap between slice and volume representations, we introduce a **Spatial Consistency Enhancement (SCE)** strategy. Unlike generic LVLMs that rely on frame sampling or key-frame stacking strategies (Xu et al., 2024; Li et al., 2024b; Huang et al., 2024), SCE performs *Volumetric Slice Composition (VSC)* by structurally combining adjacent slices along the channel dimension into locally consistent volumetric units, thereby retaining contextual spatial transitions. It further incorporates a *Tri-axial Positional Encoding (TPE)*, which injects 3D positional encodings into visual representations to enable volumetric awareness while maintaining compatibility with slice-based inputs. In addition, a *MoE Hybrid Projection (MHP)* dynamically aligns slice and volume features within a shared representation space, ensuring semantic unification with the Large Language Models (LLMs). Overall, SCE injects robust 2D/3D spatial priors while achieving a balance between efficiency and adaptability.

In clinical diagnosis, image interpretation is performed at the organ level, where observations and lesion localization are conducted within this scope (Shui et al., 2025). Building on this clinical requirement, we propose **Organ-level Semantic Enhancement (OSE)**. OSE performs *task-guided anatomical region localization*, explicitly projecting critical organ regions into the token representation space and fusing them with global visual context, thereby embedding organ-centric semantics

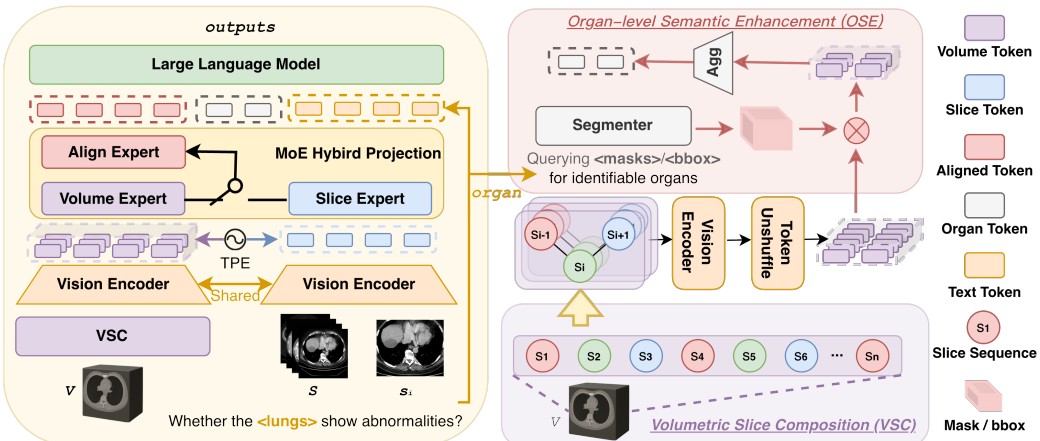

Figure 2: The architecture of OmniCT, a unified slice–volume LVLM paradigm.

into the representation. It then applies an *adaptive aggregation* to compress long-sequence representations: this mechanism preserves overall information coverage while adaptively magnifying smaller organ regions and compressing larger ones, thus highlighting the most diagnostically relevant structures. In this way, OSE explicitly incorporates region priors with high task-relevant semantic load while improving the relevance and interpretability of models in clinical tasks.

Existing medical benchmarks (Hu et al., 2024; Yue et al., 2024; Yamagishi et al., 2025) often adopt multi-modality designs to evaluate the general capability of LVLMs, yet they fall short in task alignment and clinical representativeness for CT interpretation. To address this gap, we introduce **MedEval-CT**, the first holistic evaluation framework dedicated to CT images. At the data level, *MedEval-CT-Dataset* consolidates 1.7M slice-driven and volume-driven VQA samples across 7 clinical task types, establishing the largest CT resource to date (Fig. 1 (a)). At the benchmark level, *MedEval-CT-Bench* organizes hybrid evaluations along clinical problem types and organ distributions. At the toolkit level, *MedEval-CT-Factory* standardizes input handling, feature construction, and multi-dimensional metrics, supporting statistical, semantic, and LLM-based evaluations. Collectively, MedEval-CT institutionalizes fairness and comparability in medical LVLM evaluation, while providing a scalable foundation for larger and more complex clinical scenarios.

Experimental results on multiple CT-centric benchmarks show that OmniCT achieves substantial improvements over existing methods, as illustrated by the radar chart in Fig. 1 (c), validating the effectiveness of proposed unified slice–volume modeling paradigm. Our main contributions are:

• **Unified LVLM Paradigm for CT Imaging:** Bridges the gap between slice and volume representations, injecting 3D spatial priors while retaining the efficiency of 2D alignment.

• **Representation Enhancements:** We design SCE and OSE to bridge slice–volume gaps and embed organ-centric semantics, yielding spatially coherent and clinically meaningful representations.

• **MedEval-CT:** Establishes the first holistic evaluation suite for CT imaging, augmented with 1.7M multimodal VQA samples, enabling fair, comparable, and scalable assessment of medical LVLMs.

• **Substantial Performances and Strong Baseline:** OmniCT outperforms all medical LVLMs and general LVLMs with a significant margin across multiple slice- and volume-driven CT benchmarks, establishing a strong baseline for future research towards clinical medical LVLMs.

## 2 METHODOLOGY

We propose **OmniCT**, a unified slice–volume LVLM for CT-centric understanding (Fig. 2). Unlike prior medical LVLMs restricted to either 2D slices or 3D volumes, OmniCT incorporates SCE and OSE to enable comprehensive CT representation.

## 2.1 SPATIAL CONSISTENCY ENHANCEMENT

To bridge the representational gap between slices and volumes, we propose **Spatial Consistency Enhancement (SCE)** module, which injects volumetric priors into LLM while remaining compatible with slice-driven approaches. SCE leverages *Volumetric Slice Composition (VSC)*, *Tri-Axial Positional Embedding (TPE)*, and *MoE Hybrid Projection (MHP)* to unify 2D slices and 3D volumes into the LLM space, enabling localized spatial perception, spatial position encoding, and seamless alignment of slice/volume representations within the LLM space, respectively.

**Volumetric Slice Composition.** For a 3D CT volume $\mathcal{V} \in \mathbb{R}^{D \times H \times W}$, where $D$, $H$, $W$ represent the dimensions along the z, y, and x directions, respectively, VSC structurally concatenates adjacent slices along the z axis to construct locally consistent volumetric units: $\hat{s}_i = \text{Concat}(\mathcal{V}_{3i-2}, \mathcal{V}_{3i-1}, \mathcal{V}_{3i})$ for $i = 1, \ldots, \lfloor D/3 \rfloor$, where $\hat{s}_i \in \mathbb{R}^{3 \times H \times W}$ represents a reassembled unit that preserves cross-slice spatial transitions. For independent 2D slice inputs $S = \{s_1, \ldots, s_n\}, s_i \in \mathbb{R}^{1 \times H \times W}$, we simply replicate $s_i$ along the channel axis to construct $\hat{s}_i$. In this way, both 2D slices and 3D volumes are unified as a series of reassembled units $\hat{\mathcal{S}} = \{\hat{s}_i | i \in [1, n]\}$, and $\hat{s}_i$ has a size of $3 \times H \times W$, where 3 is the channel number.

**Tri-Axial Positional Embedding.** Through volumetric slice composition, 2D slices or 3D volume are transposed into unified units $\hat{\mathcal{S}}$ of size $N_s \times 3 \times H \times W$, which are processed by a vision encoder $\phi_v(\cdot \mid \theta_v)$ with parameters $\theta_v$ to obtain patch-level visual tokens $\mathcal{F}$:

$$\mathcal{F} = \phi_v(\hat{\mathcal{S}} \mid \theta_v) = \{\phi_v(\hat{s}_1 \mid \theta_v), \ldots, \phi_v(\hat{s}_{N_s} \mid \theta_v)\} \in \mathbb{R}^{N_s \times H' \times W' \times d_v}. \quad (1)$$

Here, $H' = \frac{H}{K}$ and $W' = \frac{W}{K}$ denote the spatial dimensions before flattening the patch features, and the patch size for tokenization is $3 \times K \times K$. $N_s$ represents the number of unified reassembled units and can be regarded as a new depth dimension of reassembled units. To summarize, $N_s$ reassembled units are as inputs to generate $N_s \times 1 \times H' \times W'$ tokens with a dimension of $d_v$.

To explicitly inject global volumetric awareness, we construct sinusoidal positional encodings $P = \{P^{N_s}, P^{H'}, P^{W'}\}$ along the depth $N_s$, height $H'$, and width $W'$ dimensions of the reassembled units. This yields tokens $\mathcal{Z}$ enriched with 3D positional priors:

$$\mathcal{Z} = \mathcal{F} \oplus P = \mathcal{F} \oplus P^{N_s} \oplus P^{H'} \oplus P^{W'}, \mathcal{Z} \in \mathbb{R}^{N_s \times H' \times W' \times (d_v + d_z + d_y + d_x)}, \quad (2)$$

where $\oplus$ denotes concatenating tokens with positional encodings along the feature dimension.

**MoE Hybrid Projection.** To mitigate token explosion and reduce redundancy in the visual token representation of volumetric units for native volume input, we first perform a token-level unshuffle operation on $\mathcal{Z}$. This operation clusters spatially adjacent $m \times m$ tokens into more representations while preserving spatial relationships, resulting in newly generated token representations $\hat{\mathcal{Z}}$:

$$\hat{\mathcal{Z}} = \mathcal{U}(\mathcal{Z}), \hat{\mathcal{Z}} \in \mathbb{R}^{N_s \times (H'/m) \times (W'/m) \times [(d_v + d_z + d_y + d_x) \times m^2]}, \quad (3)$$

where $\mathcal{U}$ denotes the token-level unshuffle operation, with $m = 1$ for slice inputs to preserve original resolution. Subsequently, we employ a slice–volume Mixture of Experts Hybrid Projection (MHP) $\psi(\cdot \mid \theta_p)$ to align features with the LLM's representation space, formally expressed as:

$$\hat{\mathcal{F}} = \psi(\hat{\mathcal{Z}} \mid \theta_p = \{W_s, W_v, W_{\text{share}}\}) = W_{\text{share}} \sigma(W_s \hat{\mathcal{Z}} \cdot \mathbf{1}_{\text{slice}} + W_v \hat{\mathcal{Z}} \cdot \mathbf{1}_{\text{volume}}), \quad (4)$$

where $\sigma(\cdot)$ denotes the GELU activation function, and $\mathbf{1}_{\text{slice}}$ and $\mathbf{1}_{\text{volume}}$ are binary indicator functions that represent routing conditions for the slice and volume features, respectively (1 if the condition is satisfied, and 0 otherwise). The final tokens $\hat{\mathcal{F}}$ has a size of $L \times d_f$, where $L = N_s \times \frac{H'}{m} \times \frac{W'}{m}$ represents the total number of tokens, and $d_f$ denotes the output feature dimension of MHP, which takes an input feature dimension of $(d_v + d_z + d_y + d_x) \times m^2$.

Overall, the above SCE process generates unified CT tokens that are compatible with both slices and volumes, while embedding spatial positional awareness. These unified tokens are subsequently projected into the LLM representation space via MHP, serving as the input tokens for the LLM.

## 2.2 ORGAN-LEVEL SEMANTIC ENHANCEMENT

CT images are typically represented as high-resolution 3D volumes, often consisting of more than 150 axial slices with an in-plane resolution of $512 \times 512$, whereas lesions usually occupy only a

small and localized region. To enable clinically practical LVLMs capable of identifying abnormal features within such high-dimensional data, we introduce an **Organ-level Semantic Enhancement (OSE)** module within our unified framework, which consists of three components: anatomical region localization, semantic feature aggregation, and context fusion.

**Anatomical region localization.** Given the visual token representation $\hat{\mathcal{F}} \in \mathbb{R}^{L \times d_h}$ produced by SCE, we perform region-wise selection based on spatial priors of the target organ $o$. The organ mask is denoted as $\mathcal{M}_o \in \mathbb{R}^{D \times H \times W}$, including 117 anatomical structures, which is generated by TotalSegmentor (Wasserthal et al., 2023). This mask of $D \times H \times W$ is mapped to the token size by leveraging the scaling relationship between pixels and vision tokens, resulting in the organ-specific subset: $\hat{\mathcal{F}}_o = \hat{\mathcal{F}}[\hat{\mathcal{M}}_o]$, where $[\hat{\mathcal{M}}_o]$ denotes mask-based indexing for token selection. $\hat{\mathcal{F}}_o$ represents the selected tokens of size $L_o \times d_h$ for organ $o$ by the organ mask $\hat{\mathcal{M}}_o$.

**Adaptive organ-level feature aggregation.** Since different organs exhibit significant variation in scale and token length, directly concatenating them with text tokens can lead to severe length imbalance. To address this issue, we design a fixed-dimensional discriminative aggregation function $\text{Agg}(\cdot)$, which compresses $\hat{\mathcal{F}}_o$ into a unified size:

$$\hat{f}_o = \text{Agg}(\hat{\mathcal{F}}_o), \hat{f}_o \in \mathbb{R}^{L_c \times d_h}, \hat{\mathcal{F}}_o \in \mathbb{R}^{L_o \times d_h}, \tag{5}$$

where $L_c$ denotes the fixed number of aggregated tokens compressed from $L_o$. This aggregation not only reduces token redundancy but also introduces a "magnification effect" for small organs, enhancing fine-grained lesion features. Simultaneously, it applies a "compression effect" to large organs or global regions, effectively minimizing redundancy and preserving essential information.

Finally, the organ-level aggregated representation $\hat{f}_o$ is concatenated with the global visual tokens $\hat{\mathcal{F}}$ to generate global-local vision tokens $\hat{\mathcal{F}}_{OSE}$: $\hat{\mathcal{F}}_{OSE} = [\hat{\mathcal{F}}; \hat{f}_o]$, and combined with text tokens $\mathcal{E}$ as input to the LLM backbone, forming a semantically enhanced multimodal representation.

Overall, OSE enhances discriminative capability at the local (organ) level while maintaining contextual consistency at the global level, thus delivering more relevant and interpretable representations for downstream clinical reasoning tasks.

## 2.3 TRAINING STRATEGY

After applying Spatial Consistency Enhancement and Organ-level Semantic Enhancement, we obtain enhanced medical visual features $\hat{\mathcal{F}}_{OSE} \in \mathbb{R}^{(L+L_c) \times d_h}$. Meanwhile, the text query $Q = \{q_1, \ldots, q_m\}$ is embedded with text embedding matrix $\phi_t(\cdot|\theta_t)$:

$$\mathcal{E} = \phi_t(Q|\theta_t) \in \mathbb{R}^{m \times d_h}.$$

The two modalities are concatenated into a unified input $\mathcal{T} = [\hat{\mathcal{F}}_{OSE}; \mathcal{E}]$, which is fed into the LLM to model the conditional probability distribution. The overall optimization objective is formulated as minimizing the autoregressive cross-entropy loss:

$$\min_{\theta} \mathbb{E}_{(\mathcal{T},y)\sim\mathcal{D}} \left[ -\sum_{t=1}^{N_y} \log P(y_t \mid y_{<t}; \mathcal{T}; \theta) \right], \theta = \begin{cases} \{\theta_p\}, & \text{Pretraining Stage,} \\ \{\theta_p, \theta_{llm}\}, & \text{Instruction Tuning Stage.} \end{cases} \tag{6}$$

## 3 DATASET

Current medical benchmarks predominantly emphasize broad multi-modal capabilities, yet fall short in capturing the domain-specific demands of CT-based clinical interpretation. To bridge this gap, we introduce **MedEval-CT**, the first holistic evaluation framework for CT understanding, structured along three complementary dimensions: Datasets (*MedEval-CT-Dataset*), Benchmarks (*MedEval-CT-Bench*), and Tools (*Data Orchestration Engine* and *MedEval-CT-Factory*).

### 3.1 MEDEVAL-CT

**MedEval-CT-Dataset.** We introduce MedEval-CT-Dataset, the largest unified CT imaging resource to date, comprising over 1.7 million VQA samples from 170,280 independent 3D volumes and

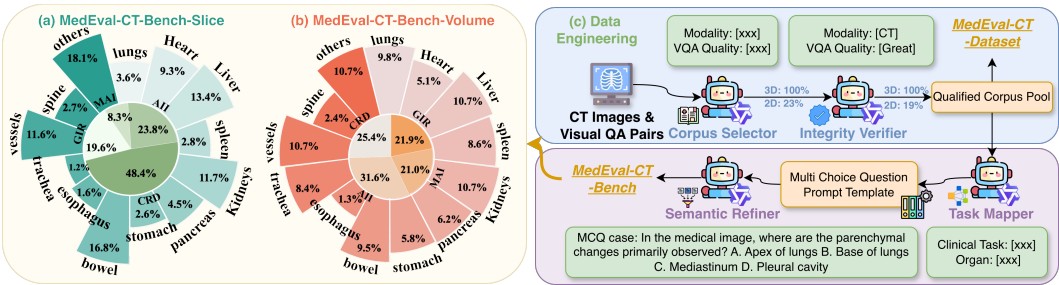

Figure 3: (a) and (b) illustrate the data distribution of MedEval-CT-Bench at the slice and volume levels, respectively, encompassing both the clinical-based categorization (4 types: GIR, MAI, AII, and CRD) and the organ-level distribution (13 organs). (c) presents the data engineering pipeline.

327,063 standalone 2D slices. The two subsets are collected from distinct, non-overlapping sources, ensuring that the 2D slices are neither extracted nor sampled from the 3D volumes. To fully leverage the rich spatial structure and clinical semantics of volumetric CT data, each 3D volume is annotated with multiple VQA instances covering diverse diagnostic perspectives, whereas the 2D subset mainly supports slice-level interpretation with typically one question per slice. Overall, the dataset enables comprehensive evaluation of both 2D clinical understanding and 3D volumetric perception, accounting for 16.3% and 83.7% of the data, respectively. As shown in Fig. 3, the dataset is systematically partitioned across three dimensions: task types, clinical categories, and organs, enabling multi-faceted evaluation of LVLMs. For task types, it spans seven medical VQA scenarios, from structured to open-ended tasks: (1) Free-form QA (38.0%) (2) Multiple Choice (18.0%) (3) Short Answers (13.7%) (4) Report Generation (11.5%) (5) Caption (7.2%) (6) Dialogue (6.2%) and (7) Judgment (5.5%). Clinical categories reflect increasing difficulty, progressing from basic anatomical recognition to expert-level reasoning: (1) General Imaging Recognition (GIR, 21.9%) (2) Medical Abnormality Identification (MAI, 23.6%) (3) Advanced Imaging Interpretation (AII, 34.5%) and (4) Clinical Reasoning and Decision (CRD, 19.9%). Organ-wise, it covers lungs (32.4%), vessels (8.2%), heart (6.6%), liver (4.9%), kidneys (4.3%), and additional regions like spine, trachea, and esophagus, ensuring robust anatomical diversity. Overall, MedEval-CT surpasses existing datasets in scale and granularity, providing high-resolution distributions across tasks, clinical expertise, and organs to advance the development of LVLMs for CT imaging. Data sources and other details are presented in Table 8 and Appendix E.

**MedEval-CT-Bench.** Based on the MedEval-CT-Dataset, we further construct MedEval-CT-Bench, the first systematic hybrid benchmark tailored for slice-volume CT. Its design emphasizes *task–organ dual balance*: on the one hand, we perform stratified sampling across different clinical problem types (GIR, MAI, AII, and CRD), ensuring full task-spectrum coverage from low-level interpretation to high-level reasoning; on the other hand, we maintain balanced organ representation, strengthening core organs (heart, lungs, liver, kidneys, etc.) while retaining long-tail structures (spine, trachea, esophagus, etc.), thereby guaranteeing fairness and comparability in clinical. To further improve clinical semantic fidelity, we propose *clinical-granularity rewriting*, which refines test questions to a more fine-grained clinical level and adds more confounding answer options while maintaining their diagnostic intent, ensuring they better reflect the variations encountered in real-world diagnostic scenarios. In summary, MedEval-CT-Bench represents significant advancements in task hierarchy, organ-level balance, and clinical authenticity, offering a more rigorous and demanding benchmark for CT understanding evaluation.

**Data Orchestration Engine.** We introduce a Data Orchestration Engine to support the construction of MedEval-CT. The engine comprises four complementary modules that collaborate across key stages, forming a self-consistent medical knowledge pipeline. It enables end-to-end capabilities for large-scale sampling, clinical consistency verification, structured task mapping, and semantic refinement: **(i) Corpus Selector:** Combines LVLM capabilities with rule-based constraints to filter CT samples from multi-source imaging datasets, ensuring representativeness across modality (2D slice/3D volume), anatomy (heart, lungs, liver, etc.), resolution, and image quality. **(ii) Integrity Verifier:** Leverages multi-modal reasoning and rule-based checks, supplemented by a 10% manual audit, to guarantee alignment between images and texts in modality, organ semantics, and pairing consistency. **(iii) Task Mapper:** Maps qualified samples to four *clinical task categories* and thirteen *organ classes*, ensuring balanced task complexity and anatomical coverage in MedEval-CT-Bench.

**(iv) Semantic Refiner:** Rewrites test questions under clinical context, introducing synonymous phrasing, terminology variations, and subtle confounding options to generate semantically similar but more discriminative multiple-choice items, thereby enhancing the benchmark's ability to evaluate clinical reasoning. Overall, the engine constructs a large-scale yet distribution-balanced MedEval-CT-Dataset while ensuring that MedEval-CT-Bench achieves reliability in terms of task hierarchy, organ balance, and clinical authenticity. Details are provided in the Appendix F.

**MedEval-CT-Factory.** We introduce MedEval-CT-Factory, an institutionalized evaluation factory designed to address the heterogeneity of inputs, features, and outputs in medical LVLMs. At the **input level**, the Factory standardizes diverse CT data formats, including DICOM, NIfTI, arrays, and slice sequences, enabling seamless 2D/3D processing. At the **feature level**, it unifies model inputs (single images, multi-slice sequences, videos, or volumes) via frame sampling, resampling, and projection strategies. At the **output level**, it provides a multi-layer evaluation protocol, ranging from statistical metrics (BLEU (Papineni et al., 2002), ROUGE (Lin, 2004)), to semantic metrics (BERTScore (Zhang et al., 2019), embedding similarity (Zhang et al., 2025b)), and further to LLM-based evaluation simulating clinical reasoning. Overall, MedEval-CT-Factory streamlines complex engineering workflows into a standardized framework, ensuring comparability across models. Serving as the fourth pillar of the MedEval-CT paradigm alongside the Dataset, Bench, and Data Engine, it is designed as a toolbox to enhance both the efficiency and fairness of LVLM evaluation in the CT domain. (The framework and details of MedEval-CT-Factory are shown in Appendix G).

## 4 EXPERIMENTS

**Baseline Comparisons.** To comprehensively evaluate the performance of OmniCT against existing open-source medical LVLMs as well as strong general-purpose LVLMs, we select a diverse set of baseline models that systematically cover both the general-to-medical spectrum and the 2D-to-3D setting. For the *2D slice-based benchmarks*, we include general-purpose LVLMs such as InternVL3 (Zhu et al., 2025), Qwen2.5-VL (Bai et al., 2025b), and GPT-5 (Wang et al., 2025), together with representative medical-domain models including HealthGPT (Lin et al., 2025), HuatuoGPT-Vision-Qwen2.5 (Chen et al., 2024b), MedGemma-4B-IT (Sellergren et al., 2025), MedVLM-R1-2B (Pan et al., 2025), Lingshu (Xu et al., 2025), and RadFM (Wu et al., 2025). For the *3D volume-based benchmarks*, we compare OmniCT with strong general LVLMs (MiniCPM-V-4.5 (Yu et al., 2025), Qwen2.5-VL, and GPT-5) as well as specialized 3D medical LVLMs, including M3D-LaMed (Bai et al., 2024), CT-CHAT (Hamamci et al., 2024a), and RadFM. The **Data Details**, **Model Details**, and **Implementation Details** are shown in Appendix E.

### 4.1 MAIN EXPERIMENTS

**Slice-driven Understanding.** We systematically evaluate OmniCT on four mainstream VQA benchmarks. As shown in the Table 2, medical LVLMs (e.g., HuatuoGPT-V-Qwen2.5, MedGemma) demonstrate relatively strong medical semantic understanding in certain tasks but remain limited in overall performance, often encountering bottlenecks on complex tasks.

Table 1: Ablation analysis of OmniCT.

| SCE | OSE | Public Bench. | | MedEval-CT-Bench | | |
|---|---|---|---|---|---|---|
| | | **2D** | **3D** | **Organ** | **Task** | **Avg.** |
| - | - | 78.68 | 62.17 | 76.51 | 78.41 | 77.62 |
| ✓ | - | 80.14 | 63.68 | 76.79 | 78.69 | 78.06 |
| - | ✓ | 80.74 | 65.37 | 77.02 | 79.42 | 78.62 |
| ✓ | ✓ | **81.45** | **66.15** | **78.24** | **80.27** | **79.62** |

In contrast, general LVLMs achieve competitive or even superior results on some benchmarks, reflecting their strengths in language reasoning, but lack adaptation to CT images. For comparison, RadFM, although capable of handling both slice and volume inputs, achieves the weakest performance across all slice benchmarks, with an average score of only 32.12, failing to meet the demands of fine-grained CT tasks. Under the same evaluation protocol, OmniCT consistently surpasses existing models at both 3B and 7B scales: the 7B version achieves an average score of **81.45**, exceeding the second-best model Lingshu by more than **+11.01**. These results demonstrate the robustness and comprehensiveness of OmniCT on slice-driven tasks.

**Volume-driven Understanding.** As shown in Table 3, we further assess OmniCT on M3D, CT-RATE, and 3D-RAD to evaluate its volumetric perception capability for CT volume. Results show that existing volume-driven medical LVLMs (e.g., M3D-LaMed-7B/4B, CT-CHAT) achieve strong performance on specific subtasks—for example, CT-CHAT reaches 86.46 on CT-RATE multi-choice—but their overall averages remain below 36, highlighting limitations in coverage and stabil-

Table 2: The comparison of **OmniCT** with other LVLMs on 2D CT benchmarks.

| Model | #Params | SLAKE | | VQA-RAD | | OmniMedVQA | | RadFig-VQA | | | Avg. |
|---|---|---|---|---|---|---|---|---|---|---|---|
| | | Close | Open | Close | Open | Task1 | Task2 | Easy | Medium | Hard | |
| *Med-LVLM (Slice-centric)* | | | | | | | | | | | |
| HealthGPT | 4B | 74.74 | 56.33 | 71.88 | 33.45 | 57.36 | 54.20 | 70.81 | 71.22 | 72.90 | 62.54 |
| HuatuoGPT-V-Qwen2.5 | 7B | 72.68 | 44.19 | 72.92 | 35.68 | 71.07 | 83.07 | 76.56 | 73.76 | 71.74 | 66.85 |
| MedGemma-4B-IT | 4B | 68.04 | 53.95 | 56.25 | 33.55 | 61.42 | 67.00 | 64.59 | 65.40 | 64.20 | 59.38 |
| MedVLM-R1-2B | 2B | - | - | - | - | 59.90 | 67.37 | 55.50 | 54.51 | 55.56 | - |
| Lingshu | 7B | 80.93 | 74.23 | 75.00 | 34.62 | 68.02 | 69.97 | 77.51 | 78.48 | 75.22 | 70.44 |
| *General-LVLM* | | | | | | | | | | | |
| InternVL3 | 8B | 73.20 | 60.88 | 69.79 | 34.79 | 63.96 | 71.78 | 68.42 | 70.46 | 68.55 | 64.65 |
| Qwen2.5-VL | 8B | 69.59 | 47.86 | 69.59 | 35.54 | 62.94 | 65.92 | 65.07 | 69.45 | 67.10 | 61.45 |
| GPT-5 | - | 78.35 | 45.86 | 70.83 | 41.05 | 67.00 | 69.10 | 80.86 | 78.90 | 81.74 | 68.19 |
| *Med-LVLM (Multi-granularity)* | | | | | | | | | | | |
| RadFM | 14B | 51.03 | 43.88 | 53.12 | 20.29 | 30.97 | 28.29 | 23.92 | 19.75 | 17.83 | 32.12 |
| **OmniCT (Ours)** | 3B | 77.84 | 85.32 | 70.83 | 30.01 | 97.46 | 97.25 | 79.43 | 82.03 | 79.13 | 77.71 |
| **OmniCT (Ours)** | 7B | **85.05** | **87.20** | **76.04** | 36.32 | **97.97** | **98.70** | **82.30** | **85.82** | **83.62** | **81.45** |

ity. General LVLMs also exhibit strong cross-modal generalization in certain volume tasks, with GPT-5 leading multiple subtasks on 3D-RAD; however, their performance is highly inconsistent and lacks domain adaptation to CT volume. By contrast, OmniCT achieves clear advantages at both 3B and 7B scales: the 3B version reaches **87.38** on CT-RATE multi-choice with an average of **63.48**, while the 7B version achieves **85.69** on the LTD task of 3D-RAD, pushing its overall average to **66.15**—significantly outperforming all compared models. In addition, considering the significant importance of CT report generation, we performed 18-class abnormality label prediction for the report generation task on CT-RATE using fine-tuned RadBERT (Yan et al., 2022). The results show that OmniCT outperforms most volume-driven CT models and previous unified models (see Table 16), and performs similarly to models specifically designed for CT volume report generation (Hamamci et al., 2024b; Di Piazza et al., 2025). This validates the superiority of OmniCT in 3D spatial modeling and cross-task consistency. Across both slice-driven and volume-driven benchmarks, OmniCT demonstrates stable and comprehensive superiority at different scales, highlighting its holistic perception of spatial–semantic features in CT volume understanding tasks.

## 4.2 ABLATION ANALYSIS

We conduct a systematic ablation study on the proposed SCE and OSE modules across multiple public 2D/3D CT benchmarks and our MedEval-CT-Bench. Notably, we keep the VSC and the MHP fixed throughout all experiments, since they serve as fundamental mechanisms for coupling 2D slice understanding with 3D volumetric perception. Under this prerequisite design, we further analyze the contribution of SCE and OSE. Results are summarized in Table 1. On the 2D slice benchmarks, the baseline achieves an average score of 79.38; introducing SCE alone improves performance to 80.14, while adding OSE alone yields 80.74. When both are combined, the performance further increases to **81.45**, achieving the best results. On the 3D volume benchmarks, the baseline starts at 62.17; adding SCE improves it to 63.68, while adding OSE alone boosts it to 65.37. The complete model combining both modules reaches the highest score of **66.15**. On the MedEval-CT-Bench, OmniCT consistently outperforms the baseline with the addition of the SCE and OSE. The improvements in both organ-level and clinical-level tasks further validate the effectiveness of these two modules. Overall, both SCE and OSE contribute significantly to performance gains, with even stronger effects observed on volume-driven tasks, demonstrating the effectiveness and complementarity of the proposed enhancements. Moreover, MHP is a crucial component for synergizing 2D and 3D modalities. We provide further analysis of its beneficial role in cross-modal generalization in Appendix H.2 (i).

## 4.3 IN-DEPTH STUDY

**(i) Performance Advantages of Mixed Data Training.** As shown in Figure 4 (a), OmniCT consistently achieves the best performance across different proportions of mixed data, demonstrating its strong adaptability to cross-modal modeling. OmniCT exhibits strong performance even under single-modality training. We attribute this behavior to the combination of a unified single-tower semantic space and the MHP, which together enable projection patterns learned from slices to extend naturally to volumes, and symmetrically allow volume-trained representations to transfer back to

Table 3: The comparison of **OmniCT** with other LVLMs on 3D CT benchmarks.

| Model | #Params | M3D | | | CT-RATE | | | 3D-RAD | | | | | Avg. |
|---|---|---|---|---|---|---|---|---|---|---|---|---|---|
| | | Cap | Close | Open | Multi choice | Clinical Entity | Report | I.O. | A.D. | E.D. | STD. | LTD. | |
| *Med-LVLM (Volume-centric)* | | | | | | | | | | | | | |
| **M3D-LaMed-7B** | **7B** | 24.79 | 75.78 | 56.09 | 47.44 | 18.15 | 16.18 | 16.85 | 16.71 | 18.00 | 25.47 | 24.17 | 30.88 |
| **M3D-LaMed-4B** | **4B** | **46.30** | 75.08 | 53.83 | 59.29 | 13.66 | 13.46 | 17.60 | 17.49 | 40.25 | 25.40 | 24.31 | 35.15 |
| **CT-CHAT** | **8B** | 21.21 | 35.88 | 21.81 | 86.46 | 49.95 | 46.76 | 31.56 | 29.98 | 45.33 | 12.95 | 13.68 | 35.97 |
| *General-LVLM* | | | | | | | | | | | | | |
| **MiniCPM-V-4_5** | **9B** | 18.44 | 43.20 | 26.89 | 69.21 | 26.21 | 23.21 | 28.03 | 29.80 | 30.98 | 12.70 | 16.32 | 29.54 |
| **Qwen2.5-VL** | **8B** | 22.62 | 48.64 | 28.99 | 61.34 | 37.51 | 26.84 | 30.51 | 30.60 | 41.28 | 9.19 | 13.05 | 31.87 |
| **GPT-5** | **-** | 21.66 | 50.36 | 33.60 | 64.27 | 34.44 | 24.86 | 32.98 | 35.22 | 67.00 | 59.07 | 77.97 | 45.59 |
| *Med-LVLM (Multi-granularity)* | | | | | | | | | | | | | |
| **RadFM** | **14B** | 22.62 | 30.39 | 19.82 | 63.93 | 19.52 | 17.92 | 23.25 | 24.67 | 29.20 | 44.11 | 42.99 | 30.77 |
| **OmniCT (Ours)** | **3B** | 27.75 | 81.24 | 62.16 | 87.38 | 63.43 | 51.67 | 52.02 | 51.43 | 84.75 | 64.43 | 72.05 | 63.48 |
| **OmniCT (Ours)** | **7B** | 26.61 | **83.84** | **63.88** | **89.80** | **63.99** | **52.48** | **53.68** | **51.97** | **87.77** | **67.91** | **85.69** | **66.15** |

slice. A more detailed analysis of this mechanism is provided in Appendix H.2 (vi). We conduct balanced sampling to verify that OmniCT can facilitate effective knowledge fusion between 2D and 3D modalities through mixed training.

**(ii) 2D Encoders *vs.* 3D Encoders.** Given the inherent differences in design objectives and input modes, directly applying 3D encoders to 2D inputs often requires artificial adaptations such as depth replication, which compromises the fairness of comparison. Therefore, we conduct evaluations in native 3D settings. As shown in Figure 4 (b), even though M3D-CLIP (Bai et al., 2024) is pretrained with contrastive learning on the M3D dataset, it does not exhibit a clear advantage over 2D encoders such as DINOv3 (245 tokens) (Siméoni et al., 2025) and SigLIP (405 tokens) (Zhai et al., 2023), despite using the largest number of visual tokens (512). These results indicate that, at this stage, 2D encoders not only provide a more natural compatibility with both 2D and 3D inputs but also demonstrate stronger generalization across tasks, organs, and modalities. To assess the generality of this finding, we additionally evaluate several recent native 3D encoders (Wan et al., 2025; Wang et al., 2023) under the same protocol; the results are provided in Table 4. It is worth noting that we are not claiming that 2D features can fully represent 3D volumes. Instead, we offer a more measured assessment: at the current stage, through structured reorganization and volume-level embedding, more generalizable 2D encoders can robustly carry 3D spatial information. This design does not collapse dimensionality; rather, it retains spatial structures and relationships that remain interpretable from a 3D perspective on top of a 2D semantic backbone.

**(iii) Organ- and Task-level Performance Analysis.** On the organ level (Figure 4 (c)), OmniCT consistently outperforms baselines across the chest, transition zone, and abdomen. The advantage is particularly striking for anatomically challenging small organs such as the pancreas and esophagus, where most existing LVLMs suffer severe performance degradation. This highlights OmniCT's ability to capture fine-grained organ semantics and boundary cues, effectively filling a critical blind spot of prior models in handling complex anatomical structures. On the task level (Figure 4 (d)), performance shows a clear gradient with respect to clinical difficulty: while most models display a significant gap between low-level anatomical recognition and high-level reasoning, OmniCT maintains consistently strong results across all levels, substantially narrowing this gap. This stability demonstrates that OmniCT not only enhance local anatomical discriminability but also reinforce consistency in clinical reasoning.

## 5 RELATED WORK

**Slice-driven Medical LVLMs.** With the success of Large Language Models in general language (Liu et al., 2024; Yang et al., 2025) and vision (Qiu et al., 2020; Bai et al., 2025a), early explorations focused on adapting general LVLM paradigms to the medical domain, such as LLaVA-Med (Li et al., 2023) and Med-Flamingo (Moor et al., 2023), which leveraged medical image–text pairs and instruction data to enable initial medical capabilities. Subsequently, a series of more general-purpose medical LVLMs emerged, including RadFM (Wu et al., 2025), BiomedGPT (Luo et al., 2023), HuatuoGPT-Vision (Chen et al., 2024a), and Lingshu (Xu et al., 2025). These models advanced the field through large-scale data curation (Bansal et al., 2024), reasoning-enhanced training strategies (Pan et al., 2025; Xu et al., 2025), multi-task generalization (Jiang et al., 2024),

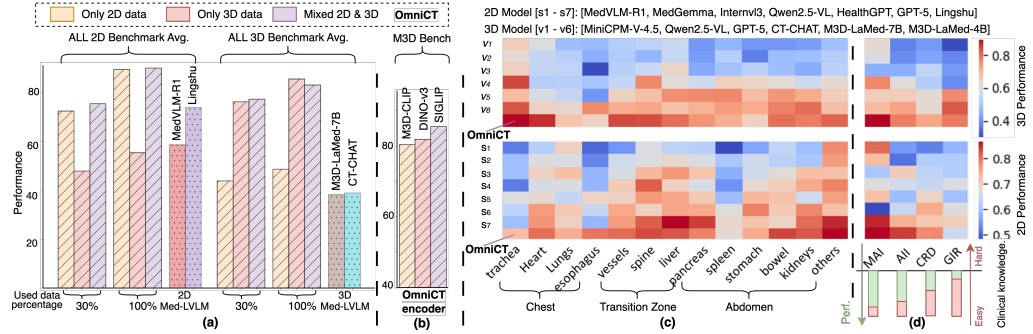

Figure 4: (a) Comparison of OmniCT with 2D/3D LVLMs on 2D/3D benchmarks using 30%, 100% training data of 2D, 3D, and mixed 2D/3D. (b) The study of using a 3D vision encoder, 2D vision encoders by different pre-training ways. (c) Per-organ performance heatmap of 2D/3D models and OmniCT on 2D/3D MedEval-CT-Bench. (d) Performance heatmaps by clinical task category and bar charts comparing performance with clinical knowledge requirements across task categories.

Table 4: 2D vs. 3D Encoder Comparison.

| Encoder | Token Budget Ratio | Plane | Phase | Organ | Abnormality | Location | Avg. |
|---------|-------------------|-------|-------|-------|-------------|----------|------|
| M3D-CLIP | 1.26× | 99.0 | 84.8 | 77.1 | 78.2 | 63.1 | 80.4 |
| VideoMAEv2 | 0.97× | 91.3 | 74.7 | 76.9 | 75.3 | 62.8 | 76.2 |
| Wan2.1-VAE | 1.42× | 97.9 | 76.4 | 77.1 | 74.7 | 63.9 | 78.0 |
| DINOv3 | 0.61× | 99.5 | 88.0 | **77.8** | 78.9 | 65.3 | 81.9 |
| SigLip | 1.00× | **99.5** | **90.2** | 78.4 | **79.2** | **67.4** | **82.9** |

and domain-specific knowledge integration (Sellergren et al., 2025). Recently, models such as CXR-LLaVA (Lee et al., 2025) and EyecareGPT (Li et al., 2025a) have demonstrated stronger adaptability and diagnostic value in modality-specific and specialty-oriented tasks (Xie et al., 2025; Hao et al., 2025). Nevertheless, despite substantial progress in data scale, architectural design, and task diversity, slice-driven medical LVLMs remain constrained by their reliance on planar inputs, limiting their ability to capture the spatial consistency and cross-slice dependencies essential for CT understanding.

**Volume-driven Medical LVLMs.** To overcome the limitations of 2D modeling, research has increasingly turned to 3D volumetric imaging, employing dedicated datasets, 3D encoders, and cross-modal alignment modules to strengthen spatial modeling in clinical tasks (Wu et al., 2025). M3D-LaMed (Bai et al., 2024) established a comprehensive evaluation system across multiple volumetric medical tasks, while CT-CHAT (Hamamci et al., 2024c) introduced paired chest CT data and an architecture tailored for fine-grained analysis and dialog-based interaction. At the methodological level, Med-2E3 (Shi et al., 2024) combined 2D and 3D encoders and enhanced reasoning consistency through dynamic cross-slice scoring, whereas Med3DInsight (Chen et al., 2024c) aligned a 3D encoder with a 2D LVLM, achieving strong performance in both segmentation and classification. Nevertheless, the lack of a unified clinical evaluation framework and efficient slice–volume collaboration mechanisms continues to limit adaptability and scalability.

## 6 CONCLUSION

We propose OmniCT, a unified slice-volume LVLM for CT analysis. Through the proposed SCE and OSE modules, OmniCT achieves spatially coherent and clinically grounded representations, leading OmniCT to realize new state-of-the-art performances on multiple benchmarks. We further propose MedEval-CT, a unified, fair, and comprehensive evaluation framework for 2D/3D CT analysis. Detailed evaluations reveal that existing general-purpose and medical LVLMs exhibit significant performance biases across clinical tasks for different organs. In contrast, OmniCT demonstrates exceptional capability with balanced performance across all organs, which will encourage LVLMs to focus on enhancing clinical capabilities for various organs in the CT domain.

## ACKNOWLEDGMENTS

This work has been supported in part by the NSFC (No. 62436007), the China Postdoctoral Science Foundation under Grant Number 2024M752794, the ZJNSF (No. LZ25F020004), the Key Research and Development Projects in Zhejiang Province (No. 2025C01128, 2025C01030, 2025C02156), Zhejiang University Education Foundation Qizhen Scholar Foundation.

## ETHICS STATEMENT

This work adheres to the ICLR Code of Ethics. No human subjects or animal experimentation were involved in this study. All medical imaging datasets used in our experiments, including SLAKE, VQA-RAD, OmniMedVQA, RadFig-VQA, M3D, CT-RATE, and 3D-RAD, are publicly available under relevant research licenses and comply with usage guidelines, ensuring no violation of privacy or ethical standards. No personally identifiable information (PII) was included, and all CT data were pre-processed (e.g., windowing, resampling, and anonymization) before use. We took special care to avoid any misuse of data and to ensure that all evaluations were conducted fairly and transparently. Our contributions focus solely on methodological and benchmarking innovations, without clinical or diagnostic decision-making implications.

## REPRODUCIBILITY STATEMENT

We have made every effort to ensure the reproducibility of our results. All codes and datasets will be released in an open repository with detailed documentation. The experimental setup, including model architectures, training hyperparameters, optimization strategies, is described in full detail in the main text and appendix. In addition, we rely on multiple widely used public datasets (e.g., SLAKE, VQA-RAD, M3D, CT-RATE) to facilitate verification and cross-comparison. Our proposed Data Orchestration Engine ensures consistent data preprocessing and evaluation, further improving reproducibility across different models. We believe these measures will enable the community to replicate our work, benchmark future models fairly, and extend the development of medical LVLMs in CT understanding.

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

# A  APPENDIX

Section B. LLM usage statement.

Section C. Notation Table.

Section D. Extended Related Work.

Section E. Implementation Details.

Section F. Data Orchestration Engine.

Section G. Mechanism of MedEval-CT-Factory

Section H. Supplementary experiments.

# B  LLM USAGE STATEMENT

In this work, we primarily employ LLMs in the following two aspects: (i) Data construction: We use LLMs for data annotation, cleaning, filtering, and rewriting, thereby building MedEval-CT-Dataset and MedEval-CT-Bench. (ii) Manuscript polishing: We leverage LLMs to review the grammar and improve the clarity and accuracy of the manuscript, ensuring that our methodology is properly and comprehensively presented. (iii) Model evaluation: We include several advanced LLMs as baseline methods for comparison. For open-weight models, we download the publicly released checkpoints and conduct inference locally. For closed-source models, we perform evaluation through their official APIs. All baseline results are obtained under a unified evaluation protocol to ensure fairness and comparability.

# C  NOTATION TABLE

To provide a comprehensive overview of the notations used throughout the paper, we present a summary of key symbols and their corresponding definitions in Table 5. This table serves as a convenient quick reference, covering the main variables and operators involved in our formulation. We hope that this notation list facilitates the understanding of our methodology and improves the readability of the paper by enabling readers to easily recall the meaning of each symbol.

# D  EXTENDED RELATED WORK

In recent times, the release of multiple Multi-modal Large Language Models (MLLMs) has driven innovations in vision-language fusion, long temporal sequence processing, and scenario adaptability (Hong et al., 2025; Team et al., 2025), laying a solid foundation for cross-domain applications. Prominent foundation models such as Qwen2.5-VL (Bai et al., 2025b), GPT-4o (Achiam et al., 2023), Claude 3.5 (AI, 2024), InternVL3 (Zhu et al., 2025), and the latest GPT-5 (Wang et al., 2025) have continuously advanced in multi-modal understanding, long-sequence processing, multi-task learning, and vertical domains like healthcare (Arora et al., 2025), demonstrating exceptional potential. These advancements are primarily driven by high-quality data curation and iterative algorithmic optimization. However, as foundation models, maintaining a balance between general-purpose capabilities and domain-specific expertise remains a significant challenge.

# E  IMPLEMENTATION DETAILS

**Data Details.** For 2D slice evaluation, we construct test sets based on SLAKE (Liu et al., 2021), VQA-RAD (Lau et al., 2018), OmniMedVQA (Hu et al., 2024), and RadFig (Yamagishi et al., 2025), where all samples are systematically filtered by the data engine (Section 3) to retain only high-quality CT VQA data. For 3D volume evaluation, we adopt existing benchmarks including M3D (Bai et al., 2024), CT-RATE (Hamamci et al., 2024a), and 3D-RAD (Gai et al., 2025) to cover the full spectrum of CT volumetric scenarios. Regarding evaluation metrics, Accuracy is used for closed-end and multiple-choice tasks, while open-ended QA is assessed by a weighted

combination of BLEU (Papineni et al., 2002), ROUGE (Lin, 2004), Token-F1 (Saab et al., 2024), and BERTScore (Zhang et al., 2019), balancing lexical matching with semantic alignment to achieve multi-level quality measurement. For data pre-processing, all CT volumes with preserved metadata are windowed to $[-1000, 1000]$ and resampled to $32 \times 384 \times 384$.

**Model Details and Implementation Details.** We use siglip-so400m-patch14-384 (Zhai et al., 2023) as the vision encoder and Qwen2.5 (Team, 2024) as the backbone LLM, with AdamW (Loshchilov & Hutter, 2017) as the optimizer. During pretraining, only the MoE Hybrid Projection is updated to perform cross-modal alignment, with a learning rate of $2 \times 10^{-4}$. In the vision instruction tuning stage, both the projection layer and LLM parameters are optimized, with the learning rate reduced to $5 \times 10^{-5}$ to ensure stable convergence. All experiments are trained with a global batch size of 256 and a warmup–cosine learning rate scheduler. Unless otherwise specified, experiments are conducted under the 7B parameter scale. The specific hyperparameter settings can be found in Table 6.

## F  DATA ORCHESTRATION ENGINE

The construction of MedEval-CT is powered by the proposed Data Orchestration Engine, which integrates four complementary modules. Specifically, the Corpus Selector and Integrity Verifier are implemented with Qwen2.5-VL-72B (Bai et al., 2025b), while the Task Mapper and Semantic Refiner leverage Qwen3-237B-A3B (Yang et al., 2025), thereby exploiting the complementary strengths of different models in large-scale data filtering and semantic refinement. The specific prompt designs for each module can be found in Fig. 5.

To ensure the reliability and independence of MedEval-CT, we proactively conduct a systematic audit of potential data overlap during its construction. For the slice datasets, we note that RO-COv2 (Ronan L.M., 2024), PubMedVision (Chen et al., 2024b), LLaVA-Med (Li et al., 2023), and RadFig-VQA (Yamagishi et al., 2025) originate from PMC-OA (Lin et al., 2023b). Although these four datasets follow their own automated curation pipelines, we further apply a two-stage dedupli-cation strategy within them: perceptual image hashing (pHash) (Monga & Evans, 2006) is used to cluster visually similar images, followed by BiomedCLIP (Zhang et al., 2023) feature matching to remove image–text pairs with high semantic similarity. For the volume datasets, we strictly adhere to the official splits of M3D (Bai et al., 2024), CT-RATE (Hamamci et al., 2024a), and 3D-RAD (Gai et al., 2025); even in cases where datasets share underlying CT volumes, we avoid any cross-dataset training or evaluation. Throughout the pipeline, we retain only modality-consistent and high-quality CT scans, filtering out blurry, artifact-heavy, or low-resolution samples. These measures allow MedEval-CT to maintain strict separation in data sourcing, partitioning, and deduplication, effec-tively minimizing the risks of training–testing contamination and data leakage.

To further address the concern that using Qwen-family models in both the data pipeline and the LLM base model might introduce family-specific bias or circularity, we additionally instantiate OmniCT with a different LLM backbone, Phi-4-mini (Abouelenin et al., 2025), while keeping the training data and optimization protocol unchanged. As shown in Table 7, OmniCT with Phi-4-mini achieves performance that is highly comparable to, and on several benchmarks slightly better than, the Qwen2.5-3B variant across both slice-driven (SLAKE, VQA-RAD, OmniMedVQA, RadFig-VQA) and volume-driven (M3D, CT-RATE, 3D-RAD, MedEval-CT-Bench) benchmarks. This con-sistency indicates that the observed gains mainly stem from the proposed unified framework itself rather than from any base model-specific preference or bias induced by the models used in the data construction pipeline.

## G  MECHANISM OF MEDEVAL-CT-FACTORY

The logical structure of MedEval-CT-Factory is illustrated in Figure 6. This section explains its design motivations and module responsibilities from a framework-level perspective.

**Unified Processing of Heterogeneous Formats.** MedEval-CT-Factory begins at the input level, where commonly used medical imaging formats are standardized. Medical data often come in di-verse forms such as DICOM, NIfTI, NRRD, 3D arrays, RGB slices, and slice sequences, which dif-fer significantly in metadata organization, spatial resolution, and storage layouts. The Factory maps

You are a medical multimodal quality control assistant. Given a medical image and its corresponding text input, determine whether this image–text pair is a valid CT dataset. Evaluate the following aspects:
1. Modality Check:
 – Confirm whether the image is a CT scan (computed tomography). Exclude non–CT images such as X–ray, MRI, ultrasound, or natural photographs.
2. Image Quality Assessment:
 – Detect any noise, motion artifacts, distortions, color shifts/pseudo–coloring, or incomplete slices that degrade diagnostic quality.
 – Verify that grayscale intensity resembles standard CT features (close to Hounsfield Unit range, without unnatural colors or abnormal contrast).
3. Text–Image Consistency:
 – Check whether the text matches the image (e.g., it should mention CT or be relevant to CT content).
 – Flag cases where the text is unrelated, mismatched, or inaccurate.
4. Overall Usability:
 – Mark as Valid only if the image is a CT scan, the quality is acceptable, and the text is relevant to the image.
 – Otherwise, mark as Invalid and briefly explain the reason (e.g., wrong modality, low quality, text–image mismatch).
Output format:
 – Validity: Valid / Invalid
 – Reason: short explanation

**Corpus Selector**

---

You are a Medical Data Integrity Verifier. The samples you receive have already passed an initial filtering step (Corpus Selector) and are considered possible CT image–text pairs. Your role is to conduct a stricter integrity audit, ensuring the data is sufficiently reliable and standardized for training high–quality medical multimodal models.
Evaluate each sample from the following perspectives:
1. Data Completeness:
 – Verify that the image is presented as a coherent slice or volume, with no missing parts or corrupted frames.
 – Check that the text description is semantically complete, not truncated, garbled, or nonsensical.
2. Data Consistency:
 – Ensure the image and text remain consistent in details (e.g., anatomical focus, scan type, common CT terminology).
 – Flag subtle mismatches (e.g., text mentions contrast agent but the image does not show contrast features).
3. Format and Standardization:
 – Confirm that the image adheres to basic medical imaging conventions (grayscale representation, consistent orientation, no artificial coloring).
 – Ensure the text is clear, medically relevant, and not casual or irrelevant.
4. Potential Flaws and Risks:
 – Identify latent issues such as mild artifacts, unusual labeling, or vague descriptions.
 – Judge whether these flaws undermine the usability of the sample.
5. Final Judgment:
 – Mark as High–Quality / Complete & Valid only if no serious flaws are detected.
 – Otherwise, mark as Risky / Low–Quality and briefly explain the issue.
Output format:
 – Integrity Check: Complete & Valid / Risky
 – Reason: short explanation

**Integrity Verifier**

---

Identify the primary organ or anatomical region mentioned in the question, and map it to one of the categories:
 1. lungs
 2. heart
 3. liver
 4. spleen
 5. kidneys
 6. pancreas
 7. stomach
 8. bowel (small_bowel, duodenum, colon)
 9. head (brain, skull)
 10. gallbladder
 11. adrenal_gland
 12. esophagus
 13. trachea
 14. thyroid_gland
 15. urinary_bladder
 16. prostate
 17. vessels
 18. muscles
 19. spine
 20. ribs_sternum
 21. upper_extremities
 22. lower_extremities
 23. others (not included above, or unclear)
Rules:
 – If multiple organs are mentioned, choose the most relevant/primary organ.
 – If the question only refers to modality or general region (not a specific organ) → choose 23. others.
 – If the question is vague or unclear → choose 23. others.
Output format:
organ_class: [1–23]

**Task Mapper**

---

Classify the question into one of the following clinical task categories:
A. General Imaging Recognition
 – Basic tasks: modality identification (CT/MRI/Ultrasound), body region classification, organ count.
B. Medical Abnormality Identification
 – Entry–level tasks: presence of abnormality, lesion, tumor, mass, abnormal density.
C. Advanced Imaging Interpretation
 – Intermediate tasks: measuring tumor size, segmentation of organ/lesion, nodule detection, structural assessment.
D. Clinical Reasoning and Decision
 – Advanced tasks: diagnostic reasoning, malignancy judgment, staging (e.g., TNM), clinical interpretation from imaging.

Rules:
 – If the question is about basic attributes → A
 – If about presence/absence of abnormality/lesion/tumor → B
 – If about quantitative analysis, segmentation, or structural features → C
 – If about diagnosis, reasoning, or staging → D

Output format:
task_class: [A–D]

**Task Mapper**

---

You are a medical data engineer tasked with rewriting medical Q&A problems into standardized four–option multiple–choice questions. Your goal is to preserve the medical logic while increasing clinical challenge through well–designed distractors.
Inputs:
 – original_question: the original question
 – original_answer: the correct answer
Steps:
1. Interpret Context
 – Ensure the rewritten question is medically accurate and consistent with the original.
2. Rewrite Question
 – Convert into a concise, clear four–option multiple–choice question.
3. Generate Distractors
 – Provide three distractors that are:
 – Relevant: connected to the medical context but incorrect.
 – Plausible: challenging enough to mislead weaker models, yet identifiable as wrong with proper reasoning.
 – Professional: use accurate medical terminology, avoid trivial errors.
Output format:
1. Return in JSON:
 {
 "question": "<Rewritten question>",
 "options": ["A. <Option A>", "B. <Option B>", "C. <Option C>", "D. <Option D>"],
 "correct_answer": "<Correct option, equivalent to original_answer in clinical semantics>"
 }
Rules:
 – Exactly one correct answer, three distractors.
 – Correct answer must match original_answer.
 – Randomize correct answer position.
 – Distractors may use similar organs, diseases, or imaging findings to increase difficulty.

**Semantic Refiner**

Figure 5: Prompt template of data orchestration engine for generating MedEval-CT.

these heterogeneous inputs into a unified representation through designated loading rules, enabling subsequent modules to perform slice-volume unified processing without relying on format-specific operations.

**Lightweight but Unified Feature Construction.** Building on the standardized inputs, the Factory provides a lightweight yet flexible feature construction layer. Instead of enforcing any model-specific preprocessing pipeline, it offers general-purpose mechanisms such as frame sampling, slice aggregation, 2D–3D projection, and resampling, allowing various LVLMs to interface with the evaluation workflow in a consistent manner.

**Multi-dimensional Evaluation Protocols.** At the output level, MedEval-CT-Factory integrates multiple evaluation strategies to accommodate the diversity of outputs produced by medical LVLMs. Rather than imposing a rigid scoring pipeline, it provides a composable and extensible evaluation space: **(i)** statistical metrics (BLEU (Papineni et al., 2002), ROUGE (Lin, 2004), METEOR (Banerjee & Lavie, 2005)) for measuring surface-level textual alignment; **(ii)** semantic metrics (BERTScore (Zhang et al., 2019), embedding similarity (Zhang et al., 2025b)) for assessing semantic correspondence; and **(iii)** LLM-based evaluation for simulating clinical reasoning, offering more qualitative judgments aligned with medical scenarios. Users may flexibly select appropriate evaluation layers according to task requirements without being restricted to a single metric.

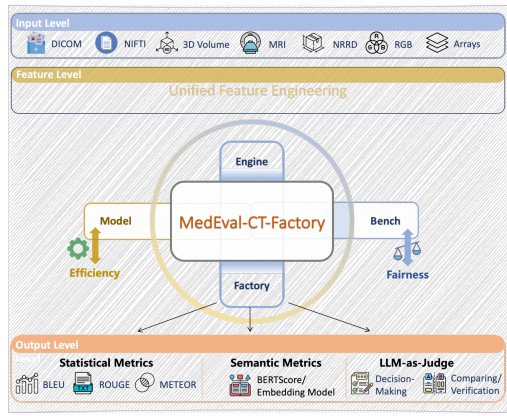

Figure 6: MedEval-CT-Factory enables standardized preprocessing and fair, consistent evaluation of medical LVLMs across benchmarks.

Overall, the Factory provides a structured, extensible, and model-agnostic framework for conducting consistent and reproducible CT LVLM evaluation. Although not all modules are used in every experiment, its modular design offers room for future extensions.

## H SUPPLEMENTARY EXPERIMENTS.

### H.1 MEDEVAL-CT-BENCH

Across both MedEval-CT-Bench-2D (Table 9) and MedEval-CT-Bench-3D (Table 10), OmniCT consistently achieves the highest overall performance, with averages of 79.80 and 77.63, respectively, surpassing strong baselines such as GPT-5-mini (OpenAI, 2025), Lingshu, CT-CHAT, and M3D-LaMed. It demonstrates robust gains across diverse organs (e.g., liver, kidneys, heart, spine) and task levels (GIR, MAI, AII, CRD), excelling particularly in advanced interpretation and reasoning. These results highlight the effectiveness of our unified slice–volume paradigm in delivering stable, cross-task generalization and comprehensive CT understanding.

### H.2 SUPPLEMENTARY ABLATION.

**(i) Analysis of Cross-Modal Generalization**. To understand the source of OmniCT's cross-modal generalization, we analyze the roles of **(i)** the unified single-tower semantic space and **(ii)** the MoE Hybrid Projection (MHP). The single-tower backbone embeds 2D slices and 3D volumes into a shared semantic neighborhood, preventing the semantic drift commonly observed in dual-encoder designs. MHP further learns a modality-adaptive mapping from visual tokens to the LLM space, allowing the projection behavior learned from 2D slices to transfer effectively to 3D representations, and vice versa. To disentangle the contributions of the two components, we compared a dual-tower without MHP configuration against the single-tower with MHP under the same training setup, and observed a substantial degradation in cross-modal generalization. The results are reported in Table 11. Therefore, these two components form a coherent mechanism that supports cross-modal

transfer: unified semantics provide a common representational anchor, and MHP supplies the flexibility needed to align slice- and volume-based tokens under a unified LVLM interface.

**(ii) t-SNE Visualization of MoE Hybrid Projection.** To further examine whether the two experts in the MHP module learn distinguishable token transformations, we project their output embeddings into a 2D space using t-SNE. As shown in Fig. 7, the features routed to the 2D expert and the 3D expert form two clearly separated clusters. This separation emerges without any explicit supervision enforcing modality-specific behavior; instead, it arises from the structural differences in the inputs (e.g., voxelized tokens with VSC/TPE for 3D vs. planar tokens for 2D) and their decoupled optimization paths before entering the shared semantic space. The visualization supports that the two experts encode modality-dependent transformations, serving the intended role of normalizing heterogeneous inputs before alignment with the LLM.

**(iii) Analysis of Organ-level Semantic Enhancement.** The OSE module leverages organ segmentation as a structural regional prior rather than a supervision target. The segmentation masks indicate organ regions with high semantic load for typical CT-based reasoning, from which OSE aggregates a compact set of discriminative tokens, while all global tokens are preserved in the feature stream. In this way, OSE explicitly strengthens organ-level semantics without sacrificing global context. Since the module relies on organ-level structural consistency instead of pixel-level boundary fitting, the high stability of TotalSegmentor in thoracoabdominal CT (average Dice 94.3% (Wasserthal et al., 2023)) is well suited for providing such regional cues. To evaluate the effectiveness of OSE and to rule out potential bias introduced by the segmentation model, we designed three alternative strategies: (i) removing ROI regions, (ii) random ROI pooling, and (iii) directly concatenating native ROI tokens. As shown in Table 12, removing ROIs yields the expected performance drop; random pooling brings limited gains mainly due to weak alignment effects arising from repeated visual tokens; and direct concatenation of native ROI tokens produces variable-length sequences that prevent stable semantic compression and offer no performance benefit. In contrast, OSE's fixed-dimensional adaptive aggregation preserves global information coverage while emphasizing diagnostically critical regions, making it better suited to the structured requirements of medical image analysis.

**(iv) Ablation of Adaptive Feature Aggregation.** We further examined the impact of different 2D/3D aggregation token settings $(m_{2D}, m_{3D})$ on model performance (Table 13). The results show that moderate aggregation (e.g., $m_{2D} = 81, m_{3D} = 90$) consistently improves both 2D and 3D performance compared to the baseline without OSE. As the aggregation context continues to grow, the gains diminish and eventually decline, likely due to excessive semantic overlap with global features that disperses the model's effective visual attention. Overall, these observations indicate that an appropriately sized set of aggregated tokens can effectively enhance organ-level semantics, increase the information density of visual tokens, and maintain a favorable balance between accuracy and computational cost.

**(v) Robustness of MedEval-CT-Bench to Answer Leakage** To reduce the risk that models exploit language artifacts instead of visual evidence, MedEval-CT-Bench's multiple-choice questions are constructed with a clinical-granularity refiner that rewrites prompts using synonymous expressions, refines clinical wording, and injects stronger distractor options. This preserves the underlying diagnostic intent while weakening template-like phrasing, simple co-occurrence patterns, and answer-position biases. We further conduct two stress tests on MedEval-CT-Bench: (i) an image–question mismatch setting, where questions are randomly paired with incorrect CT scans/volumes, and (ii) a noise substitution setting, where images are replaced by noise. As shown in Table 14, both 2D (6-way choice, random $\approx 16.7\%$) and 3D (4-way choice, random $\approx 25\%$) accuracies drop sharply toward near-random levels under mismatch/noise, while remaining high with normal inputs.

**(vi) Unified Representation Gains.** To further examine the feasibility and utility of using a 2D encoder as the semantic backbone for incorporating 3D spatial cues, we conduct a balanced sub-sampling study across slice-driven and volume-driven data. Specifically, we perform controlled ablations using 25%, 50%, and 100% of the available samples for each modality (results in Table 15). Across all settings, joint training consistently yields measurable performance gains. These results indicate that, under the current scale of available pretraining resources, 2D encoders exhibit more mature semantic generalization and thus serve as a reliable representational anchor for constructing 3D inputs. With structured spatial injection, the unified representation acquires effective volumetric awareness, enabling synergistic improvements across both slice- and volume-level tasks.

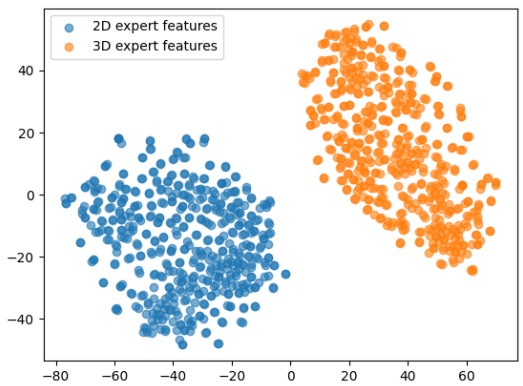

Figure 7: t-SNE plot showing distinct clusters of 2D and 3D expert features after MHP module.

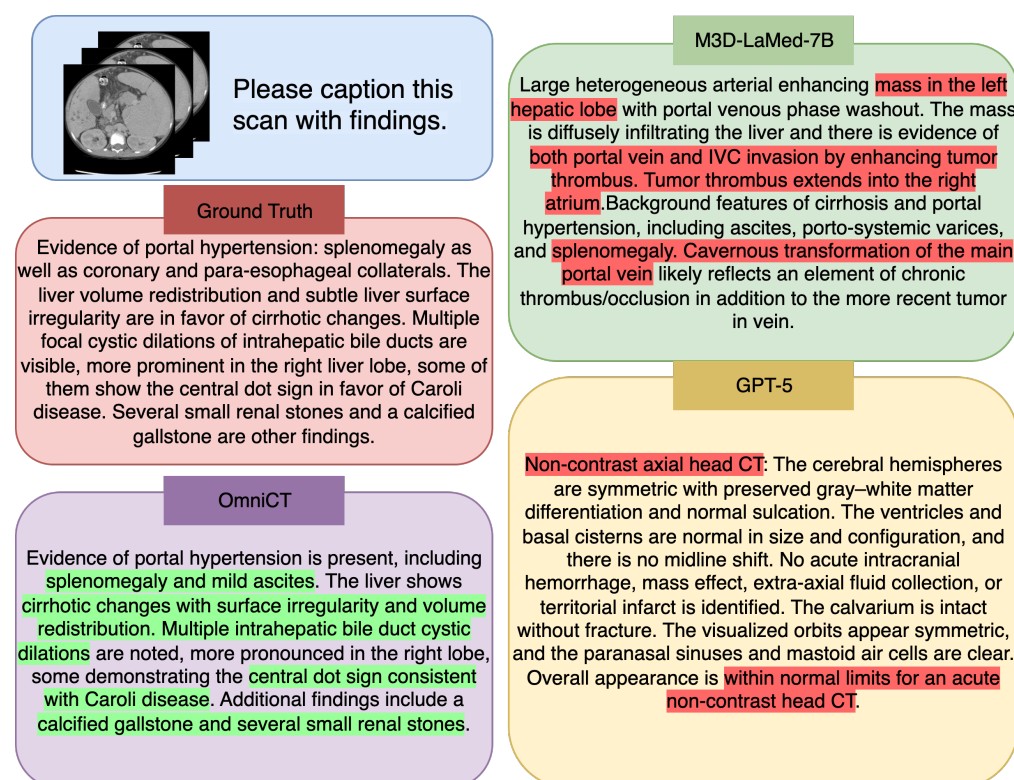

Figure 8: A case study comparing CT findings generated by different medical LVLMs with the clinical ground truth.

### H.3 CASE STUDY

In Figure 8, the Ground Truth highlights portal hypertension, cirrhotic changes, intrahepatic bile duct cystic dilatations with the central dot sign, and additional renal and gallstones. M3D-LaMed-7B, while mentioning hepatic and portal abnormalities, overemphasizes tumor invasion with incorrect findings, and GPT-5 completely deviates with irrelevant head CT descriptions. In contrast, OmniCT accurately captures the key clinical features—portal hypertension, splenomegaly, mild ascites, cirrhosis, Caroli disease signs, and urinary/gallstones—closely matching the Ground Truth with clinically coherent language, demonstrating its superior spatial–semantic consistency in chest–abdominal CT interpretation.

Table 5: Notations used throughout this paper.

| Notation | Description |
|---|---|
| $\mathcal{V} \in \mathbb{R}^{D \times H \times W}$ | 3D CT volume with depth $D$, height $H$, and width $W$ |
| $\mathcal{V}_j \in \mathbb{R}^{1 \times H \times W}$ | $j$-th 2D slice extracted from the 3D volume |
| $s_i \in \mathbb{R}^{1 \times H \times W}$ | Independent 2D slice input |
| $S = \{s_1, \ldots, s_n\}$ | Collection of independent 2D slice inputs |
| $\hat{s}_i = \text{Concat}(\mathcal{V}_{3i-2}, \mathcal{V}_{3i-1}, \mathcal{V}_{3i})$ | Reassembled volumetric unit |
| $\hat{\mathcal{S}} = \{\hat{s}_i \mid i \in [1, n]\}$ | Set of all reassembled slice units |
| $N_s$ | Number of reassembled units (new depth dimension) |
| $\phi_v(\cdot \mid \theta_v)$ | Vision encoder with parameters $\theta_v$ |
| $\mathcal{F} \in \mathbb{R}^{N_s \times H' \times W' \times d_v}$ | Patch-level visual tokens extracted from $\hat{\mathcal{S}}$ |
| $H' = \frac{H}{K}, W' = \frac{W}{K}$ | Spatial resolution of patch features after tokenization |
| $K$ | Patch size (stride along spatial dimensions) |
| $d_v$ | Dimension of visual tokens |
| $P = \{P^{N_s}, P^{H'}, P^{W'}\}$ | Sinusoidal positional encodings along depth, height, width |
| $\mathcal{Z}$ | Tokens enriched with tri-axial positional priors |
| $d_z, d_y, d_x$ | Feature dimensions of depth/height/width positional encodings |
| $\mathcal{U}$ | Token-level unshuffle operation |
| $m$ | Window size for unshuffle ($m$=1 for slice input) |
| $\hat{\mathcal{Z}}$ | Token representations after unshuffle |
| $\psi(\cdot \mid \theta_p)$ | Slice–volume MoE Hybrid Projection function |
| $\theta_p = \{W_s, W_v, W_{\text{share}}\}$ | Parameters of MoE Hybrid Projection |
| $W_s, W_v, W_{\text{share}}$ | Slice-specific, volume-specific, and shared projection matrices |
| $\mathbf{1}_{\text{slice}}, \mathbf{1}_{\text{volume}}$ | Binary indicator functions for slice/volume routing |
| $\sigma(\cdot)$ | GELU activation function |
| $\hat{\mathcal{F}} \in \mathbb{R}^{L \times d_f}$ | Projected tokens aligned with LLM space |
| $L = N_s \frac{H'}{m} \frac{W'}{m}$ | Total number of projected tokens |
| $d_f$ | Output feature dimension of MoE Hybrid Projection |
| $\mathcal{M}_o \in \mathbb{R}^{D \times H \times W}$ | Organ mask for organ $o$ (from TotalSegmentor, 117 structures) |
| $\hat{\mathcal{M}}_o$ | Organ mask mapped to token resolution |
| $\hat{\mathcal{F}}_o \in \mathbb{R}^{L_o \times d_h}$ | Subset of tokens selected by $\hat{\mathcal{M}}_o$ |
| $L_o$ | Number of tokens belonging to organ $o$ |
| $\text{Agg}(\cdot)$ | Organ-level feature aggregation function |
| $\hat{f}_o \in \mathbb{R}^{L_c \times d_h}$ | Aggregated tokens for organ $o$ |
| $L_c$ | Fixed number of aggregated tokens after compression |
| $\hat{\mathcal{F}}_{OSE}$ | Global-local vision tokens after OSE fusion |
| $Q = \{q_1, \ldots, q_m\}$ | Input text query sequence |
| $\phi_t(\cdot|\theta_t)$ | Text embedding function with parameters $\theta_t$ |
| $\mathcal{E} \in \mathbb{R}^{m \times d_h}$ | Text token embeddings |
| $\mathcal{T} = [\hat{\mathcal{F}}_{OSE}; \mathcal{E}]$ | Unified multimodal input sequence |
| $y = (y_1, \ldots, y_m)$ | Target output sequence |
| $P(y_t \mid y_{<t}; \mathcal{T}; \theta)$ | Conditional probability distribution from LLM |
| $\mathcal{D}$ | Training data distribution |
| $\theta_{llm}$ | Parameters of the LLM backbone |

Table 6: Overview of the hyperparameter settings used for training OmniCT-3B and OmniCT-7B across two stages.

| Hyperparameter | OmniCT-3B | | OmniCT-7B | |
|---|---|---|---|---|
| | Stage-1 | Stage-2 | Stage-1 | Stage-2 |
| Optimizer | AdamW | AdamW | AdamW | AdamW |
| Learning Rate of Adapter | 2e-4 | 5e-5 | 2e-4 | 5e-5 |
| Learning Rate of LLM | - | 5e-5 | - | 5e-5 |
| Global Batch Size | 256 | 256 | 256 | 256 |
| Weight Decay | 0 | 0 | 0 | 0 |
| LR Scheduler | warm up-cosine | warm up-cosine | warm up-cosine | warm up-cosine |
| Warmup Ratio | 0.03 | 0.03 | 0.03 | 0.03 |
| Epoch | 2 | 1 | 2 | 1 |
| Max Sequence Length | 2048 | 2048 | 2048 | 2048 |

Table 7: Performance of OmniCT with different LLM backbones on 2D and 3D CT benchmarks.

| Base Model | #Params | SLAKE | VQA-RAD | OmniMed VQA | RadFig -VQA | M3D | CT-RATE | 3D-RAD | MedEval-CT -Bench |
|---|---|---|---|---|---|---|---|---|---|
| Qwen2.5-3B | 3B | **81.6** | 50.4 | **97.4** | 80.2 | **57.1** | **67.5** | 64.9 | 75.9 |
| Phi-4-mini | 3.8B | 81.1 | **55.3** | 96.9 | **84.4** | 56.4 | 67.4 | **66.9** | **76.2** |

Table 8: Summary of the datasets included in MedEval-CT, with their task types, training/test sizes, sources, and licenses.

| Dataset | Task Type | Train Size | Test Size | Source | License |
|---|---|---|---|---|---|
| **Volume-driven Datasets** | | | | | |
| M3D-CAP | Report | 116065 | 100 | Radiopaedia | Apache-2.0 |
| M3D-VQA | Multiple Choice
Short Answer | 240929
240929 | 5000
5000 | Radiopaedia | Apache-2.0 |
| CT-RATEv2 | Report
Free-form QA | 93822
693760 | 6076
24149 | Istanbul Medipol University,
Mega Hospital | CC-BY-NC-SA-4.0 |
| 3D-RAD | Short Answer
Judgment
Multiple Choice | 9709
100170
26316 | 4692
23472
5746 | Istanbul Medipol University,
Mega Hospital | CC-BY-NC-SA-4.0 |
| **Slice-driven Datasets** | | | | | |
| ROCOv2 | Caption | 18663 | - | PMC-OA | Apache-2.0 |
| PubMedVision | Caption
Dialogue | 113142
112649 | -
- | PMC-OA | Apache-2.0 |
| RadFig-VQA | Multiple Choice | 46696 | 2084 | PMC-OA | CC-BY-NC-SA-4.0 |
| OmniMedVQA | Multiple Choice | 14230 | 1579 | Part of 73 datasets | CC BY & Apache-2.0 |
| LLaVA-Med | Dialogue | 10622 | - | PMC-15M | Apache-2.0 |
| MEDPIX-ClinQA | Caption
Dialogue | 3895
3895 | -
- | MEDPIX 2.0 | Apache-2.0 |
| VQA-RAD | Short Answer
Judgment | 1040
1248 | 61
96 | MEDPIX | CC0 |
| SLAKE | Short VQA
Judgment | 2598
2280 | 234
194 | Medical Segmentation Decathlon,
ChestXray-NIHCC, CHAOS | CC-BY-4.0 |

Table 9: MedEval-CT-Bench-2D

| Model | Qwen2.5-VL | InternVL3-8B | RadFM | Lingshu | HealthGPT | MedVLM-R1 | MedGemma-4B | GPT-5-mini | Ours |
|---|---|---|---|---|---|---|---|---|---|
| lungs | 72.50 | 66.25 | 18.99 | 70.00 | 63.75 | 70.00 | 66.25 | 71.25 | 74.68 |
| heart | 69.76 | 65.85 | 5.39 | 77.07 | 68.29 | 61.95 | 66.83 | 76.47 | 80.39 |
| liver | 76.95 | 70.17 | 10.20 | 86.44 | 75.25 | 62.71 | 68.47 | 75.59 | 82.31 |
| spleen | 57.38 | 54.10 | 10.00 | 68.85 | 59.02 | 47.54 | 60.66 | 67.21 | 68.33 |
| kidneys | 72.48 | 75.19 | 9.34 | 80.62 | 65.12 | 59.30 | 60.47 | 78.29 | 82.10 |
| pancreas | 72.00 | 66.00 | 8.08 | 84.00 | 71.00 | 62.00 | 66.00 | 75.00 | 78.79 |
| stomach | 64.91 | 68.42 | 7.14 | 71.93 | 73.68 | 56.14 | 64.91 | 78.95 | 75.00 |
| bowel | 67.84 | 70.27 | 9.76 | 69.46 | 72.16 | 63.51 | 62.43 | 71.08 | 83.47 |
| esophagus | 58.33 | 58.33 | 8.57 | 58.33 | 69.44 | 50.00 | 52.78 | 63.89 | 80.00 |
| trachea | 59.26 | 51.85 | 7.69 | 59.26 | 66.67 | 51.85 | 59.26 | 62.96 | 73.08 |
| vessels | 65.88 | 70.98 | 7.87 | 76.47 | 69.80 | 54.51 | 65.10 | 75.20 | 80.71 |
| spine | 73.33 | 81.67 | 6.78 | 81.67 | 76.67 | 66.67 | 66.67 | 70.00 | 83.05 |
| others | 72.25 | 78.75 | 9.27 | 85.25 | 76.00 | 78.00 | 76.50 | 72.25 | 86.47 |
| GIR | 67.75 | 72.16 | 13.95 | 75.41 | 66.36 | 64.97 | 67.75 | 72.56 | 70.70 |
| MAI | 67.03 | 72.53 | 8.29 | 87.91 | 76.92 | 84.07 | 77.47 | 56.04 | 88.95 |
| AII | 75.62 | 69.71 | 9.73 | 76.76 | 65.90 | 62.29 | 59.05 | 69.90 | 86.83 |
| CRD | 69.14 | 71.11 | 6.95 | 78.33 | 74.67 | 60.23 | 67.82 | 79.15 | 81.78 |
| Average | 68.38 | 68.43 | 9.29 | 75.75 | 70.04 | 62.10 | 65.20 | 71.52 | 79.80 |

Table 10: MedEval-CT-Bench-3D

| Model | Qwen2.5-VL | MiniCPM-V-4.5 | CT-CHAT | M3D-LaMed-Phi-3-4B | M3D-LaMed-Llama-2-7B | GPT-5-mini | Ours |
|---|---|---|---|---|---|---|---|
| lungs | 52.07 | 47.66 | 52.07 | 54.27 | 62.26 | 53.72 | 67.22 |
| heart | 54.45 | 59.16 | 51.83 | 58.64 | 63.87 | 54.45 | 86.91 |
| liver | 55.14 | 55.39 | 60.40 | 75.94 | 67.42 | 62.91 | 80.20 |
| spleen | 51.88 | 45.94 | 64.38 | 76.88 | 74.06 | 48.44 | 71.25 |
| kidneys | 47.87 | 41.60 | 55.89 | 71.68 | 80.20 | 44.11 | 78.20 |
| pancreas | 41.38 | 40.52 | 56.47 | 76.72 | 70.69 | 43.53 | 75.00 |
| stomach | 36.92 | 40.65 | 62.62 | 68.22 | 71.50 | 52.80 | 64.02 |
| bowel | 47.32 | 40.28 | 60.56 | 74.93 | 77.75 | 47.61 | 80.00 |
| esophagus | 44.68 | 46.81 | 48.94 | 40.43 | 76.60 | 29.79 | 63.83 |
| trachea | 61.41 | 65.27 | 84.57 | 68.81 | 82.96 | 64.31 | 89.39 |
| vessels | 50.88 | 51.63 | 54.14 | 64.16 | 71.68 | 52.13 | 84.46 |
| spine | 45.98 | 52.87 | 77.01 | 67.82 | 72.41 | 55.17 | 83.91 |
| others | 51.63 | 48.12 | 61.15 | 64.91 | 68.42 | 52.63 | 81.45 |
| GIR | 45.83 | 39.34 | 46.45 | 75.98 | 80.88 | 47.92 | 77.82 |
| MAI | 61.89 | 64.32 | 76.34 | 66.75 | 70.08 | 51.15 | 86.06 |
| AII | 46.40 | 45.29 | 63.10 | 67.26 | 73.28 | 51.06 | 77.18 |
| CRD | 49.47 | 48.00 | 56.01 | 63.92 | 64.45 | 58.12 | 72.78 |
| Average | 49.72 | 48.99 | 60.70 | 66.90 | 72.27 | 51.17 | 77.63 |

Table 11: Ablation of MoE Hybrid Projection.

| Training Strategy | Perf. 2D | Perf. 3D | Avg. |
|---|---|---|---|
| SigLip + M3D-CLIP (w/o MHP) | 34.57 | 30.58 | 32.58 |
| SigLip + Siglip (w/ MHP) | **55.30** | **48.61** | **51.96** |

Table 12: Ablation analysis of adaptive organ-level feature aggregation.

| ROI Strategy | Perf. 2D | Perf. 3D | Avg. |
|---|---|---|---|
| No OSE | 78.68 | 62.17 | 70.43 |
| OSE w/ native ROI | 78.37 | 62.24 | 70.31 |
| OSE w/ random ROI | 80.13 | 64.22 | 72.18 |
| OSE w/ adaptive ROI | **81.45** | **66.15** | **73.80** |

Table 13: Ablation of the OSE aggregation ratios for 2D and 3D tokens. $m_{2D}$ and $m_{3D}$ denote the numbers of aggregated organ-level tokens for 2D slices and 3D volumes, respectively.

| $m_{2D}$ | $m_{3D}$ | Perf. 2D | Perf. 3D | Avg. |
|---|---|---|---|---|
| 0 | 0 | 78.68 | 62.17 | 70.43 |
| 36 | 40 | 80.66 | 63.81 | 72.24 |
| 81 | 90 | **81.45** | **66.15** | **73.80** |
| 144 | 160 | 81.23 | 66.04 | 73.64 |
| 225 | 250 | 80.64 | 65.48 | 73.06 |

Table 14: Organ-level accuracy on MedEval-CT-Bench under normal, image–question mismatch, and noise settings

| Organ | 2D Normal | 2D Mismatch | 2D Noise | 3D Normal | 3D Mismatch | 3D Noise |
|---|---|---|---|---|---|---|
| lungs | 74.7 | 23.1 | 19.5 | 67.2 | 29.4 | 27.3 |
| heart | 80.4 | 24.0 | 14.9 | 86.9 | 31.8 | 20.1 |
| liver | 82.3 | 19.9 | 20.9 | 80.2 | 30.5 | 25.8 |
| spleen | 68.3 | 13.7 | 17.4 | 71.3 | 28.4 | 29.0 |
| kidneys | 82.1 | 25.1 | 21.1 | 78.2 | 22.3 | 22.6 |
| pancreas | 78.8 | 24.9 | 15.2 | 75.0 | 30.2 | 23.4 |
| stomach | 75.0 | 21.7 | 14.4 | 64.0 | 19.0 | 22.7 |
| bowel | 83.5 | 22.2 | 19.4 | 80.0 | 30.1 | 27.9 |
| esophagus | 80.0 | 20.3 | 18.0 | 63.8 | 28.6 | 29.2 |
| trachea | 73.1 | 18.0 | 20.6 | 89.4 | 21.8 | 24.1 |
| vessels | 80.7 | 23.9 | 19.8 | 84.5 | 30.1 | 28.6 |
| spine | 83.1 | 17.2 | 17.8 | 83.9 | 28.9 | 17.1 |
| others | 86.5 | 22.9 | 14.6 | 81.5 | 29.7 | 27.0 |

Table 15: Ablation study of unified representation gains. Compared to 2D-only and 3D-only training, mixed training consistently improves performance across all data scales while preserving the same 2D/3D ratio.

| Training Strategy | Ratio | SLAKE | VQA-RAD | RadFig-VQA | M3D-VQA | CT-RATE | 3D-RAD |
|---|---|---|---|---|---|---|---|
| 2D-Only | 25% | 70.6 | **62.5** | 77.2 | - | - | - |
| 3D-Only | 25% | - | - | - | 65.8 | 84.5 | 65.4 |
| **Mixed** | 25% | **72.2** | **62.5** | **78.2** | **69.9** | **84.9** | **66.9** |
| 2D-Only | 50% | 75.8 | 66.7 | 79.4 | - | - | - |
| 3D-Only | 50% | - | - | - | 72.7 | 86.2 | 67.2 |
| **Mixed** | 50% | **77.3** | **70.8** | **79.8** | **73.6** | **87.1** | **68.1** |
| 2D-Only | 100% | 81.0 | **72.3** | 78.2 | - | - | - |
| 3D-Only | 100% | - | - | - | 74.4 | **86.6** | 68.6 |
| **Mixed** | 100% | **81.2** | 71.8 | **81.9** | **74.7** | **86.6** | **69.3** |

Table 16: Performance of 18 types of anomaly label prediction.

| Model | Precision | Recall | F1 |
|---|---|---|---|
| RadFM | 13.1 | 6.4 | 7.2 |
| M3D-LaMed-7B | 8.1 | 2.5 | 3.5 |
| M3D-LaMed-4B | 16.5 | 8.4 | 9.6 |
| CT-CHAT | 24.3 | 38.8 | 27.2 |
| CT2Rep | 41.6 | 38.1 | 36.7 |
| CT-AGRG | 37.8 | **55.4** | **42.1** |
| OmniCT | **41.7** | 36.5 | 36.3 |

