# OpenReview forum: "OmniCT: Towards a Unified Slice-Volume LVLM for Comprehensive CT Analysis"
_ICLR.cc/2026/Conference — ICLR 2026 Poster_

### Official Review · Reviewer_GKxu · 2025-10-27

**Soundness:** 3
**Presentation:** 2
**Contribution:** 3
**Rating:** 6
**Confidence:** 3

**Summary:**

The authors propose OmniCT, a unified LVLM that handles both slice-level and volume-level CT understanding. This is achieved through two key modules: Spatial Consistency Enhancement (SCE) and Organ-level Semantic Enhancement (OSE). SCE comprises volumetric slice composition, tri-axial positional embeddings, and a slice/volume MoE hybrid projection. OSE utilizes organ masks (generated using TotalSegmentator) to localize and analyze 117 anatomical structures. Subsequently, adaptive token aggregation and fusion with global tokens are employed. Additionally, the authors introduce MedEval-CT, a comprehensive dataset, benchmark, and toolkit suite for CT. MedEval-CT boasts approximately 1.7 million slice/volume VQA samples across seven distinct task types, for organ- and task-balanced evaluation. Extensive experiments were conducted across various public 2D and 3D CT benchmarks, including SLAKE, VQA-RAD, OmniMedVQA, RadFig-VQA, M3D, CT-RATE, and 3D-RAD. The results demonstrate that OmniCT (3B/7B) outperforms prior general and medical LVLMs.

**Strengths:**

- The proposed SCE/OSE design directly addresses the gap between slice-driven detail sensitivity and volume-driven spatial reasoning, which are squarely motivated by clinical reading patterns.

- OmniCT shows consistent improvements over some models across diverse 2D and 3D CT benchmarks, with well-documented ablations supporting the claims.

- Table 1 (SCE/OSE ablation) and studies on mixed-data training, encoder choices (2D vs 3D), and organ/task-level heatmaps add useful insight and support the design claims.

- Volumetric Slice Composition + Tri-Axial Positional Embedding adds volumetric awareness while keeping slice compatibility. The MoE Hybrid Projection is a pragmatic way to route slice vs volume tokens.

**Weaknesses:**

- While MedEval-CT is positioned as “largest” (~1.7M) and “holistic,” the data sources, licensing, and deduplication across training vs evaluation (and vs existing public benchmarks) are not detailed enough to rule out contamination/leakage.

- The pipeline leans on large models (Qwen2.5-VL-72B, Qwen3-237B-A3B) for selection/mapping/refinement. This raises bias propagation questions and potential circularity if similar model families are used in training/eval.

- The 2D-encoder-dominant findings are interesting, but the native-3D comparisons seem limited (e.g., M3D-CLIP vs DINOv3/SigLIP). Stronger baselines (recent 3D ViTs / 3D CLIP variants with tuning matched to your token budgets) would make the '2D encoders suffice' claim more convincing.

- The presentation of the work, particularly the figures, could be enhanced by using more aesthetically pleasing boxes or eliminating the shadows to improve readability.

**Questions:**

- Could you provide a clear breakdown of MedEval-CT’s data sources, licenses, and how overlap with public evaluation benchmarks (e.g., SLAKE, VQA-RAD, CT-RATE) is prevented? Details on your deduplication or leakage-checking strategy would greatly strengthen the dataset’s credibility.

- Could you report sensitivity analyses on the token unshuffle parameter m and its computational trade-offs? How does varying m impact memory usage, inference latency, and accuracy for 2D vs 3D settings?

- Since OSE relies on TotalSegmentator masks, how robust is OmniCT to imperfect or noisy organ segmentations? Have you tested performance degradation under synthetic noise or partial-mask conditions?

- The results suggest 2D encoders outperform 3D ones for volumetric reasoning. Could you add or discuss comparisons against stronger native-3D models (e.g., VideoMAE-3D, Uni3D-ViT) with matched token budgets to solidify this claim?

---

> ### Author Response · Authors · 2025-11-20
> **Author Response to Reviewer GKxu (Part 1/3)**
>
> Thank you for the reviewer’s positive assessment and constructive suggestions. The reviewer notes that our work effectively bridges consistency modeling between slices and volumes, and demonstrates strong performance through extensive experiments. Below, we provide detailed responses to the reviewer’s questions.
>
> ---
>
> ### **(w1) Dataset Provenance & Leakage Control**
>
> We appreciate the reviewer’s attention to dataset provenance. All data used in the MedEval-CT-Dataset come from publicly licensed datasets, with their sources fully listed in **`Table 7`** of the manuscript and **licensing details added in the revised version**. To address concerns about contamination from shared origins, we apply explicit isolation and semantic diversification strategies during dataset construction.
>
> > 2D Datasets
>
> According to our data audit, only ROCOv2, PubMedVision, LLaVA-Med, and RadFig-VQA originate from PMC-OA, but each is produced through an independent automated generation pipeline (including rewriting, template conversion, and instruction formatting) and relies on different LLMs for annotation. When constructing MedEval-CT-Dataset, we use only RadFig-VQA as part of the evaluation set, while all other PMC-OA–related datasets are used exclusively for training to avoid cross-source coupling. All remaining 2D datasets come from entirely different data sources.
>
> > 3D Datasets
>
> M3D and CT-RATE come from different institutions; although CT-RATE and 3D-RAD share CT volumes, both are built strictly on official splits, which we follow without any cross-dataset training or cross-evaluation.
>
> In addition, MedEval-CT undergoes systematic quality filtering and deduplication during construction: we retain only CT-modality samples and remove blurred, ghosted, non-standard view, color-shifted, or low-resolution images. For the four PMC-OA–related 2D datasets, we apply pHash clustering to group near-duplicate images, followed by BiomedCLIP filtering to remove image–text pairs with high similarity in both modalities. Approximately 19% of candidate samples remain, maximizing independence and diversity across datasets.
>
> Taken together, MedEval-CT-Dataset applies strict isolation in data sourcing, splitting, and deduplication, effectively preventing train–test contamination and leakage.
>
> ---
>
> ### **(w2) Data/Model circularity**
> We fully understand the reviewer’s concern. To minimize the risk of introducing family-specific biases from the rewriting model at the data level, we use it only for paraphrasing, not for eliciting or injecting its internal knowledge. During data orchestration, we constrain the rewriting process with controlled prompts, ensuring that it generates equivalent expressions and distractor options strictly within the semantic neighborhood of the correct answer, thereby avoiding the introduction of additional prior patterns.
>
> We further test whether OmniCT exhibits any model dependence or circular bias by using LLMs from different model families as the backbone. The results show that both backbone variants exhibit highly consistent performance across all 2D/3D benchmarks, and each maintains the original performance lead:
>
> |Base Model|\#Params|SLAKE|VQA-RAD|OmniMedVQA|RadFig-VQA|M3D|CT-RATE|3D-RAD|MedEval-CT-Bench|
> |-|:-:|:-:|:-:|:-:|:-:|:-:|:-:|:-:|:-:|
> |Qwen2.5-3B|3B|81.6|50.4|97.4|80.2|57.1|67.5|64.9|75.9|
> |Phi-4-mini|3.8B|81.1|55.3|96.9|84.4|56.4|67.4|66.9|76.2|
>
> This consistency indicates that OmniCT’s performance gains arise from the framework design and data quality itself. And we appreciate the reviewer’s high-quality questions. The clarifications for **`(w1,w2)`** are added to **`Section F`** for complete reference, and we thank the reviewer for giving us the opportunity to address potential misunderstandings.

---

> ### Author Response · Authors · 2025-11-20
> **Author Response to Reviewer GKxu (Part 2/3)**
>
> ### **(w3) 2D Encoder vs. 3D Encoder**
> We thank the reviewer for requesting a deeper investigation into encoder performance. We include stronger 3D baselines and perform a systematic comparison under the same token budget and the same two-stage training strategy (M3D-Cap → M3D-VQA), covering recent representative models such as VideoMAEv2 and Wan2.1-VAE. The results are shown in the table below; despite comparable or even higher token costs, these 3D encoders do not surpass the 2D baseline across three subtask categories—organ recognition, abnormality interpretation, and spatial localization:
>
> |**Encoder**|**Token Budget Ratio**|**Plane**|**Phase**|**Organ**|**Abnormality**|**Location**|**Avg.**|
> |-|:-:|:-:|:-:|:-:|:-:|:-:|:-:|
> |SigLip|$1.00\times$|99.5|90.2|78.4|79.2|67.4|82.9|
> |VideoMAEv2|$0.97\times$|91.3|74.7|76.9|75.3|62.8|76.2|
> |Wan2.1-VAE|$1.42\times$|97.9|76.4|77.1|74.7|63.9|78.0|
>
> These results indicate that, under comparable token budgets and training strategies, current 3D encoders do not demonstrate superior generalization compared to large-scale 2D pretrained models. The 2D encoder benefits from richer semantic supervision and finer-grained local features, which yields more stable performance on clinical VQA tasks. **How to organize and exploit these fine-grained cues within a unified framework remains the direction with the greatest potential impact.**
>
> We do not rule out the emergence of stronger 3D encoders in the future, but at present, the assumption that "3D must outperform 2D" is not supported by empirical evidence. The corresponding comparisons are added to **`Section 4.3 (ii)`** in the revised manuscript, and we thank the reviewer for prompting this valuable experimental extension.
>
> ---
>
> ### **(w4) Figure presentation**
> We appreciate the reviewer’s suggestions regarding figure presentation. After reviewing the submitted version, we guess that the visible shading in several figures results from OpenReview’s PDF compression, which amplifies the light, hand-drawn–style background textures in the originals. To remove visual noise and improve readability, we replace all affected figures in the revised version with cleaner, shadow-free graphical styles. If the reviewer has additional layout or presentation recommendations, we are happy to further refine the figures.
>
> ---
>
> ### **(q1) Dataset Provenance & Leakage Control**
> We thank the reviewer for the attention to dataset provenance. This concern is essentially aligned with the points raised in **`(w1)`**. We address it fully in that section and update the relevant information in Appendix Table 7 of the revised manuscript.
>
> ---
>
> ### **(q2) Sensitivity Analysis of the Token Unshuffle Parameter $m$**
> We appreciate the reviewer’s attention to the sensitivity of $m$ and the computation–performance trade-off under 2D/3D settings. The design motivation for $m$ is to address the order-of-magnitude difference in token counts between CT volumes and slices: 3D inputs produce thousands of visual tokens, which introduce substantial gradient imbalance. Therefore, we apply mild compression ($m_{2D}$ = 1, $m_{3D}$ = 3), which keeps the token budget manageable while avoiding excessive degradation of 3D spatial structure.
>
> Following the reviewer’s suggestion, we perform a sensitivity analysis using more aggressive compression ratios:
>
> |$\mathbf{m_{2D}}$|$\mathbf{m_{3D}}$|**Token Length**|**GPU Mem**|**Infer Latency**|**2D Perf.**|**3D Perf.**|**Avg.**|
> |:-:|:-:|:-:|:-:|:-:|:-:|:-:|:-:|
> |1|3|~ 940|~ 84G|5.7 iter/s|78.7|62.2|70.5|
> |3|9|~ 250|~ 73G|7.1 iter/s|75.4|55.2|65.3|
>
> Higher compression ratios provide some efficiency gains but lead to clear performance degradation. We observe two key reasons for this:
> (i) although token unshuffle preserves lossless reordering, collapsing larger spatial blocks into single tokens increases the difficulty of cross-scale structural recognition;
> (ii) under high compression, the projection layer must increase its dimensionality substantially (adding roughly 800 MB of parameters), which contradicts the design goal of a lightweight projection module and significantly weakens 2D/3D semantic alignment quality.
>
> Therefore, ($m_{2D}$ = 1, $m_{3D}$ = 3) is not a mere tuning choice, but the trade-off point that balances efficiency, gradient stability, and semantic recoverability. Both higher and lower compression ratios lead to unacceptable performance declines at the current stage.

---

> ### Author Response · Authors · 2025-11-20
> **Author Response to Reviewer GKxu (Part 3/3)**
>
> ### **(q3) Robustness of OSE**
> We appreciate the reviewer for raising this important question regarding OSE’s stability. OSE does not treat organ segmentation as a supervision signal; instead, it uses segmentation as a region-level prior to trace and aggregate tokens with higher semantic load while always retaining the full global context. As a result, it is naturally insensitive to local mask errors.
>
> To verify this directly, we construct two types of perturbations in Table 11 of the manuscript:
> **(i) Random ROI regions:** these disrupt organ boundaries while preserving region size;
> **(i) Removal of all ROI tokens:** this simulates completely missing or severely incorrect segmentation.
>
> Both settings represent worst-case scenarios equivalent to the reviewer’s concern. The results show that model performance decreases only moderately and remains stable, indicating that OSE does not rely on precise pixel-level segmentation and instead benefits from region-level structural priors.
>
> We also further analyze the effect of aggregation rates, and observe that OSE’s adaptive aggregation yields optimal and stable gains in both 2D and 3D settings:
>
> |$\mathbf{m_{2D}}$|$\mathbf{m_{3D}}$|**Perf. 2D**|**Perf. 3D**|**Avg.**|
> |:-:|:-:|:-:|:-:|:-:|
> |0|0|78.7|62.2|70.4|
> |36|40|80.7|63.8|72.2|
> |81|90|81.5|66.2|73.8|
> |144|160|81.2|66.0|73.6|
> |225|250|80.6|65.5|73.1|
>
> In summary, OSE’s robustness arises from its design, which relies weakly on geometric priors and strongly on structured global representation. Its performance does not degrade significantly in the presence of segmentation errors. These additional clarifications are incorporated into **`Section H.2 (iii,iv)`** of the revised manuscript.
>
> ---
>
> ### **(q4) 2D Encoder vs. 3D Encoder**
> We appreciate the reviewer’s continued attention to the encoder-related conclusions. This issue is addressed systematically in **`(w3)`**, where we supplement our analysis with comparisons against stronger native 3D encoders under the same token budget and the same two-stage training strategy. The complete results are included in **`Table 13`** of the revised appendix. Across all comparable settings, 3D encoder still do not outperform the large-scale 2D encoder, further reinforcing our observation that 2D representations serve as a strong and reliable foundation for the unified framework.
>
> ---
>
> **`Summary:`** We sincerely thank the reviewer once again for the careful evaluation and valuable feedback. We provide detailed, point-by-point responses to every concern and revise the paper comprehensively based on these constructive insights. We genuinely hope that these clarifications address the reviewer’s doubts and encourage a more favorable assessment of the work’s novelty and overall quality.

---

> ### Author Response · Authors · 2025-11-26
> **Kind Check for Any Remaining Concerns**
>
> Dear Reviewer,
>
> I hope you are doing well. As the discussion window is nearing its end, we wanted to reach out to confirm whether you have any additional concerns or points for us to address. We would like to make sure that every aspect of your feedback is fully addressed before the period concludes.
>
> Your thoughtful suggestions have significantly strengthened our work, and we are happy to clarify anything further if needed.
>
> Thank you again for your careful review and continued participation.
>
> Best regards,
>
> The Authors

---

### Official Review · Reviewer_xwuZ · 2025-10-29

**Soundness:** 2
**Presentation:** 1
**Contribution:** 3
**Rating:** 4
**Confidence:** 3

**Summary:**

The authors propose OmniCT and MedEval-CT. OmniCT is a new architecture, aimed at allowing to combine slice and volumetric information in vision-encoders, allowing their LVLM to be applied to both, 3D and 2D data. Additionally, they introduce Spatial Consistency Enhancement (SCE) and Organ-level semantic enhancement (SCE).
Regarding the MedEval-CT Dataset and Benchmark, they unify the existing 2D and 3D VLM benchmarks and unify their evaluation.

**Strengths:**

The paper exceeds their baselines across many benchmarks, yielding performance improvements and pushing the state-of-the-art of medical image understanding.
Moreover they introduce a way to leverage both, 2D and 3D medical images.

**Weaknesses:**

My main concern with the paper two-fold:

Firstly and most-importantly, the motivation of the paper is unclear to me. The authors motivation originates largely from the fact that there exist 2D and 3D CT medical image datasets, however by default all CT images are 3D. Subsequently, i don't know (and the authors don't motivate well) why 2D slices are needed to be integrated. How would one get the 2D slice selection? If a professional is in the loop, why would I need the LVLM? This question puts the authors method as well as the unification of their benchmark in question.

Secondly, clarity. The paper is squeezing a lot of content into the 9 pages, which leads to a lot of important context missing or being deferred somewhere else. E.g. details on the large dataset the authors propose (MedEval-CT-Dataset) are hard to comeby. Maybe it's a collection of all already publicly available datasets, maybe it's new data? All they provide is 1.7 million 2D slices. This glossing over important details is prevalent in the paper.


Other minor concerns are:
1. The idea of leveraging 2D encoders and packing slices into RGB channels is not novel with an old example being [1] and joint training existing as well in [2].
2. The idea of leveraging organ-masks to aggregate embeddings is not new either, as it was done by fVLM before already [3].

[1] Draelos, Rachel Lea, et al. "Machine-learning-based multiple abnormality prediction with large-scale chest computed tomography volumes." Medical image analysis 67 (2021): 101857.
[2] Xie, Yutong, et al. "Unimiss: Universal medical self-supervised learning via breaking dimensionality barrier." European Conference on Computer Vision. Cham: Springer Nature Switzerland, 2022.
[3] Shui, Zhongyi, et al. "Large-scale and Fine-grained Vision-language Pre-training for Enhanced CT Image Understanding." The Thirteenth International Conference on Learning Representations.

**Questions:**

Q1: MedEval-CT-Dataset: Where does this data come from? 1.7 million samples means what in Volumes/Slices?
Q2: MedEval-CT-Bench: Why is a joint benchmark necessary? Why not just use the existing slice-wise benchmarks and the existing 3D benchmarks?
Q3: Did the authors evaluate how LLMs are able to solve the VQA (in particular the multiple-choice) questions? There is work that shows that question design alone can leak information on the correct answers.
Q4: Did you evaluate quantitative metrics? I.e. CT-RATE report generation has a RadBERT classifier to yield abnormality labels, which would circumvent the potential VQA question leakage.

---

> ### Author Response · Authors · 2025-11-20
> **Author Response to Reviewer xwuZ (Part 1/3)**
>
> Thank you for reviewing our work and for the thoughtful feedback. We appreciate the reviewer’s acknowledgment of our method’s strong benchmark performance and the value of leveraging unified slice–volume medical imaging. Below, we respond to each of the reviewer’s specific concerns.
>
> ---
>
> ### **(w1) Motivation clarification**
> We thank the reviewer for highlighting the need for clearer motivation. Our proposed unified paradigm is designed to embed shared 2D/3D semantics within a single representation space, reflecting the inherent interplay between 2D and 3D data forms, model design requirements, and clinical workflows. Notably, **`reviewers RrqG and GKxu`** found our motivation strong and well justified, yet we fully respect and value your concern. Below, we elaborate more directly on: why 2D/3D unification is needed in medical imaging, why this direction is not an artificially constructed task, and why we consider it essential to advance this paradigm in medical LVLMs.
>
> First, from the perspective of data and annotation structure, although CT is inherently 3D, medical imaging–text corpora have long exhibited a pronounced 2D bias. Resources such as PMC-OA, MedTrinity-25M, and large volumes of PACS reports are predominantly curated in a "key slice + description" format rather than with volume-level annotations. Thus, integrating 2D information is not an artificial problem setup-**it is a necessary accommodation of the vast and mature 2D semantic resources embedded in medical corpora.**
>
> Second, from the standpoint of model evolution and semantic continuity, medical LVLMs originated almost entirely from 2D semantic alignment, and their semantic spaces are naturally organized around 2D imagery. Building an isolated system exclusively for 3D inputs would **create a semantic discontinuity** and **prevent reuse of established capabilities.**
>
> Finally, from the perspective of clinical workflow, **diagnostic practice inherently alternates between 2D and 3D reasoning:** clinicians rely on 3D context to assess anatomical relationships while repeatedly revisiting key slices for fine-grained interpretation and measurement. The selection of these slices varies across tasks and lesion types, making it inherently non-standardizable. An LVLM capable of understanding both slices and volumes can surface candidate evidence for clinician verification, shifting the workflow from slice-by-slice searching to targeted review—reducing burden and improving consistency without replacing professional judgment.
>
> Moreover, recent representative works have also treated unified modality support as a core design objective [1,2], and explicitly identify the current separation of 2D and 3D modeling as a limiting factor. These independent developments further substantiate the practical need for unified 2D/3D handling in medical foundation models.
>
> Taken together, we firmly believe that unified representation is a necessary prerequisite for medical LVLMs to achieve transferability, scalability, and real clinical utility. We also believe that OmniCT’s unified paradigm provides a solid foundation for cross-modal modeling in medical LVLMs and will continue to benefit subsequent research and practice.
>
> [1] Wu, C., et al. "Towards generalist foundation model for radiology by leveraging web-scale 2d&3d medical data," in Nature Communications, vol. 16, no. 1, pp. 7866, 2025.
>
> [2] Jiang, S., et al. "Omniv-med: Scaling medical vision-language model for universal visual understanding," in arXiv preprint arXiv:2504.14692, 2025.
>
> ---
>
> ### **(w2) Details clarification**
> We thank the reviewer for raising concerns regarding the allocation of space and the clarity of dataset details. Due to the 9-page limit of the main manuscript, we prioritized presenting task types and anatomical distributions of dataset in the main text, and moved fully verifiable data-source details to Table 7 in the appendix. Regarding the reviewer’s reference to **`1.7M 2D slices`**, we clarify that this number denotes the total volume of slice-driven and volume-driven VQA samples.
>
> In the revised manuscript, we strengthened the cross-references between the main text and the appendix in **`Section 3.1: MedEval-CT-Dataset`**. Our balance between space constraints and presentation depth was intended to maintain a clear narrative flow in the main body while ensuring that all necessary details remain fully reproducible in the appendix.
>
> If any additional clarification would further assist the reviewer’s evaluation, we would be happy to provide it promptly.

---

> ### Author Response · Authors · 2025-11-20
> **Author Response to Reviewer xwuZ (Part 2/3)**
>
> ### **(w3) Relation to earlier work**
> We understand the reviewer’s concerns about these potential overlaps. Our targeted responses are as follows.
>
> > Packing slices and joint training
>
> Earlier works that use multi-channel slice inputs primarily address shape compatibility or pseudo-3D representations; they **do not establish a consistent geometric coordinate system** that simultaneously accommodates both 2D and 3D inputs. OmniCT, by contrast, uses volumetric slice composition (VSC) and tri-axial positional embedding (TPE) to explicitly map 2D slices and 3D volumes into a voxel-isomorphic space, after which MoE hybrid projection (MHP) aligns these visual features with the LLM backbone’s semantic space. Multi-slice fusion in our framework serves only as a backward-compatible input option rather than the source of unification.
>
> Moreover, joint training itself is not a new concept—it is a standard strategy in multi-task and multi-modal learning. The key question is **how modalities are jointly modeled and optimized.** OmniCT’s single-tower architecture and differentiated feature flows achieve consistent cross-modal alignment. Notably, our experiments are the first to observe that, under a unified geometric and semantic space, a 2D encoder surpasses 3D-pretrained encoders on multiple 3D tasks (Fig. 4(b)), validating the effectiveness of the unified alignment mechanism itself.
>
> > Difference from fvlm
>
> fvlm uses organ masks as explicit semantic supervision during upstream pre-training to construct anatomy–text alignment. In contrast, OSE operates at the LVLM input stage and uses no supervision signal. It treats organ regions only as lightweight spatial indicators and performs structured tracing and aggregation of global visual tokens within the already unified voxel coordinate space, aiming to increase semantic density and local discriminability rather than to achieve semantic alignment.
>
> The two methods **differ clearly in training objectives, usage stages, and the semantic state of the input features.** Therefore, they are not comparable nor functionally interchangeable.
>
> ---
>
> ### **(q1) MedEval-CT-Dataset**
> We list the data sources and scales of the MedEval-CT-Dataset systematically in **`Table 7`** of the manuscript, and in the revised version we provide the original sources and licensing information for each component.
>
> We supplement the key statistics here: after rigorous filtering and construction, the MedEval-CT-Dataset contains 1.7M VQA samples, including 170,280 CT volumes (flattened into 20M+ slices) and 327,063 standalone CT slices.
>
> ---
>
> ### **(q2) MedEval-CT-Bench**
> We appreciate the reviewer’s question regarding the necessity of a unified benchmark. We clarify that the goal of MedEval-CT-Bench is not to evaluate cross-modal joint reasoning, but to provide a CT-focused evaluation framework with **more complete task coverage, more balanced organ distribution, and clinical expressions** that more closely reflect real-world use. The 2D-VQA and 3D-VQA subsets can be used entirely independently, without any dependence on cross-modal joint settings.
>
> Our motivation for introducing this benchmark is that existing 2D and 3D CT benchmarks differ substantially in task types, organ distributions, and semantic granularity, making it difficult to form a comparable evaluation space. They lack consistency in task difficulty, organ bias, and clinical realism. As a result, current benchmarks only capture fragments of model capability and cannot systematically assess a model’s stability across different organs and reasoning difficulty.
>
> MedEval-CT-Bench is designed systematically to address the above gaps:
> **(1) Task–organ dual balancing:** it performs stratified sampling across four core clinical tasks (GIR/MAI/AII/CRD) while maintaining balanced distributions between major organs (heart, lung, liver, kidney) and long-tail structures (spine, esophagus).
> **(2) Clinical-grade rewriting:** it rewrites samples with finer clinical granularity and introduces more challenging distractor options, making the evaluation closer to the ambiguity and variability encountered in real diagnostic scenarios.
>
> Therefore, MedEval-CT-Bench serves as a more comprehensive, balanced, and clinically stringent benchmark for CT understanding.

---

> > ### Comment · Reviewer_xwuZ · 2025-11-21
> > **Reponse Part 2**
> >
> > ### Differences to fVLM
> >
> > > OSE operates at the LVLM input stage and uses no supervision signal
> >
> > To me it seems like fVLM and the way OSE uses the masks is very similar, as in aggregating the tokens within the semantic mask. Albeit fVLM applies it to image-text alignment and this paper uses it for a different training objective.
> >
> > > Notably, our experiments are the first to observe that, under a unified geometric and semantic space, a 2D encoder surpasses 3D-pretrained encoders on multiple 3D tasks (Fig. 4(b))
> >
> > I disagree. The experiments provided do not allow to make this claim. The authors do not train an encoder in isolation and evaluate their 2D encoder against 3D encoders. To do so it would have been needed to isolate the effects of the encoder from the effects of the LVLM training. E.g. it would have been necessary to use a pre-trained 3D encoder (e.g. CT-CLIP or some vision-only 3D MAE) and then use its embeddings and train a LVLM in the same fashion on that, and on the authors' proposed encoder.
> >
> > ### Q1) Dataset details
> >
> > Thanks for the clarification. _This information should not be part of the appendix but communicated in the main._
> > In particular, because this __provides important context on all the experimental results__, since it means the model is applied in-distribution and is not generalizing to these benchmarks.
> >
> > ### Q2) Need for joint benchmark
> >
> > Thanks for clarifying.

---

> ### Author Response · Authors · 2025-11-20
> **Author Response to Reviewer xwuZ (Part 3/3)**
>
> ### **(q3) VQA evaluation**
> We do conduct systematic evaluations of the model on multiple-choice VQA tasks, covering both public 2D/3D benchmarks and our MedEval-CT-Bench. In response to the reviewer’s concern about potential question leakage, we clarify the following:
>
> First, the multiple-choice questions in public benchmarks are long-established community settings whose structures have been validated across many prior studies. Second, in MedEval-CT-Bench, we apply semantic rewriting through the clinical-granularity refiner in our data orchestration engine: it rewrites each question via synonym substitution, refinement of clinical semantics, and insertion of more challenging distractor options. This reduces the influence of templated wording, statistical co-occurrence, and answer-position bias.
>
> We further include two control experiments in MedEval-CT-Bench—image–question mismatch and noise substitution—where questions are randomly paired with the wrong CT images, or the input image is replaced with noise. Under both conditions, model accuracy drops sharply to near-random levels:
>
> |**Organ**|**2D Normal**|**2D Mismatch**|**2D Noise**|**3D Normal**|**3D Mismatch**|**3D Noise**|
> |-|:-:|:-:|:-:|:-:|:-:|:-:|
> |lungs|74.7|23.1|19.5|67.2|29.4|27.3|
> |heart|80.4|24.0|14.9|86.9|31.8|20.1|
> |liver|82.3|19.9|20.9|80.2|30.5|25.8|
> |spleen|68.3|13.7|17.4|71.3|28.4|29.0|
> |kidneys|82.1|25.1|21.1|78.2|22.3|22.6|
> |pancreas|78.8|24.9|15.2|75.0|30.2|23.4|
> |stomach|75.0|21.7|14.4|64.0|19.0|22.7|
> |bowel|83.5|22.2|19.4|80.0|30.1|27.9|
> |esophagus|80.0|20.3|18.0|63.8|28.6|29.2|
> |trachea|73.1|18.0|20.6|89.4|21.8|24.1|
> |vessels|80.7|23.9|19.8|84.5|30.1|28.6|
> |spine|83.1|17.2|17.8|83.9|28.9|17.1|
> |others|86.5|22.9|14.6|81.4|29.7|27.0|
>
> Therefore, the reported performance reflects the model’s genuine clinical reasoning ability under visual–language conditions, rather than any leakage effect arising from question format or linguistic patterns. We add this clarification to **`H.2 (v)`** for completeness, and we thank the reviewer for prompting this valuable extension.
>
> ---
>
> ### **(q4) Quantitative metric evaluation**
> We appreciate the reviewer’s suggestion regarding quantitative metric. We clarify that RadBERT is not a general-purpose evaluator in CT-RATE: its role is to extract abnormality labels from radiology reports to construct the training label space, not to serve as a comparator for open-ended report generation.
>
> Therefore, we follow the mainstream LVLM evaluation protocol: using accuracy for closed-form and multiple-choice questions, and BLEU/ROUGE/Token-F1/BERTScore for open-ended and report-generation tasks. To further strengthen evaluation robustness, we additionally introduce LLM-as-judge (gpt-5) to assign a 0–5 semantic consistency score for generated reports, providing a semantic-level complement to traditional n-gram metrics.
>
> The results show that the LLM scores follow the same trend as related measures, and collectively indicate that OmniCT consistently outperforms existing methods across diverse generation tasks:
>
> |**Model**|**M3D(CAP)-LLM**|**M3D(CAP)**|**CT-RATE(Report)-LLM**|**CT-RATE(Report)**|
> |-|:-:|:-:|:-:|:-:|
> |M3D-LaMed-7B|3.42|24.79|1.61|16.18|
> |CT-RATE|3.18|21.21|3.69|46.76|
> |RadFM|2.96|22.62|2.47|24.86|
> |OmniCT-7B|3.91|26.61|4.11|52.48|
>
> ---
>
> **`Summary:`** Once again, we thank the reviewer for the careful assessment and constructive feedback. We address each concern with specific, reproducible responses in this rebuttal. Based on these clarifications and the added empirical evidence, we respectfully invite the reviewer to reassess the technical contributions and experimental support of this work, and to consider adjusting the current score accordingly.

---

> > ### Comment · Reviewer_xwuZ · 2025-11-21
> > **Response Part 3**
> >
> > ### Q3 VQA evaluation
> >
> > Thanks for the additional experiment! I think this is a really cool showcase to show that the Language itself does not leak the correct responses and increases confidence in the results and benchmark and IMO quality of the paper!
> >
> > ### Q4 Quantitative metric evaluation
> >
> > I appreciate the effort here and partially agree about the open-endedness, but the RadBERT classifier still adds value:
> > Instead of measuring open-endedness, it allows measuring the subset of abnormalities that are mentioned in the report. So while it may not be sensitive to __everything__ it still adds value. In particular it allows putting the values into perspective with other papers that report these values.
> > _Moreover, I want to emphasize that I __do not__ need the classification values to be state-of-the-art! This model is not aimed at this and this expectation would be unfair. If the authors provide me the RadBERT classifier scores I will raise my scores __independent of them being bold or not__._

---

> > > ### Author Response · Authors · 2025-11-26
> > > **Kind Check for Any Remaining Concerns**
> > >
> > > Dear Reviewer,
> > >
> > > I hope you are doing well. With the discussion phase approaching its last few days, we would like to kindly check whether there are any remaining issues you would like us to clarify or expand upon. We aim to ensure that all your comments have been handled carefully before the discussion closes.
> > >
> > > Your constructive feedback has greatly helped us refine the manuscript, and we welcome any further observations you may wish to share.
> > >
> > > Thank you for your time and dedication throughout this process.
> > >
> > > Best regards,
> > >
> > > The Authors

---

> ### Comment · Reviewer_xwuZ · 2025-11-21
> **Response Part 1**
>
> ### Regarding W1
> I am not convinced by the response:  _"diagnostic practice inherently alternates between 2D and 3D reasoning"_ This point only shows the limitations of humans. If we could observe the entire 3D volume in a 3D way, clinicians would probably do it. Moreover, in 3D medical imaging, it is well established that 3D native models are superior to 2D ones, quite famously in nnU-Net [1], where 3D encoders outperform 2D encoders consistently.
>
> The _"key slice + description"_ argument is not something that I find convincing either. The key-slice is selected __after__ understanding what is going on in the image. I.e., the clinician checked everything and then selected a representative slice as localisation. It's not a verification that reformatting 3D data as single slices is the right way of interpreting things.
>
> If the argument is made that the 2D data is available, so it should be used, then I would like to see verification that the inclusion of the 2D data strengthens the capabilities of the encoder to reason in 3D.
>
>  Moreover, the point of __"semantic discontinuity"__  and __"prevent reuse of established capabilities"__ is tied closely to the point raised above, that if 2D images don't help 3D understanding, it makes no sense to try to stay __"continuous"__ or to unify the 2D and 3D domains. Generally, it is to note that there exist various ways of trying to leverage 2D data for 3D understanding. BIUD [2] leverages X-Rays with Reports to improve CT understanding, Merlin [3] inflates 2D weights to 3D kernels. The question I raise is not related to "Does it make sense to benefit from 2D data" my question raised is "Does it make sense to represent 3D data as 2D" and as a consequence "Is it beneficial to have a joint 2D and 2.5D vision encoder."
>
>
> ### Regarding W2:
>
> Thanks for clarifying this to some extend. However, it would be nice to further disentangle this a bit: How many original pure images (2D and 3D) were involved? I.e. one could create 100 questions on a single 2D slice or 3D volume, or one could create 1 question per 2D slice or 3D volume.
>
> [1]: Isensee, Fabian, et al. "nnU-Net: a self-configuring method for deep learning-based biomedical image segmentation." Nature methods 18.2 (2021): 203-211.
>
> [2]: Cao, Weiwei, et al. "Bootstrapping chest ct image understanding by distilling knowledge from x-ray expert models." Proceedings of the IEEE/CVF Conference on Computer Vision and Pattern Recognition. 2024.
>
> [3]: Blankemeier, Louis, et al. "Merlin: A vision language foundation model for 3d computed tomography." Research Square (2024): rs-3.

---

> ### Author Response · Authors · 2025-11-22
> **Author Response to Reviewer xwuZ - Round 2 (Part 1/2)**
>
> > Regarding W1
>
> We sincerely appreciate the reviewers’ constructive feedback and would like to establish clearer consensus on several points.
>
> From a technical standpoint, we fully agree on the potential advantages of 3D-native perceptual models. In fact, our design considerations stem from a practical technical reality: in medical imaging scenarios, 3D semantic encoders pretrained via CL/SSL/MAE paradigms still struggle to surpass large-scale supervised 3D models [1]. **This is precisely why the 3D encoders in our experiments do not exhibit significant advantages, and it also explains the strong performance of supervised models (e.g., nnU-Net).** Due to the closed-domain nature of medical data, current semantic-level 3D encoders still require more favorable conditions to realize their full potential.
>
> To make this conclusion more transparent in our work, we conducted a unified comparison—using identical training protocols (3D datasets + 3D baselines)—across multiple categories of 3D encoders:
>
> |**Encoder**|Training Paradigm|**Token Budget Ratio**|**Plane**|**Phase**|**Organ**|**Abnormality**|**Location**|**Avg.**|
> |-|:-:|:-:|:-:|:-:|:-:|:-:|:-:|:-:|
> |SigLip|CL|$1.00\times$|99.5|90.2|78.4|79.2|67.4|82.9|
> |DINOv3|Self-Distillation SSL|$0.61\times$|99.5|88.0|77.8|78.9|65.3|81.9|
> |VideoMAEv2|MAE|$0.97\times$|91.3|74.7|76.9|75.3|62.8|76.2|
> |Wan2.1-VAE|VAE|$1.42\times$|97.9|76.4|77.1|74.7|63.9|78.0|
> |M3D-CLIP|CL|$1.26\times$|99.0|84.8|77.1|78.2|63.1|80.4|
> |CT-CLIP|CL|$1.42\times$|98.3|85.7|76.2|76.4|64.2|80.2|
>
> Consistent with our analysis, the 3D encoders do not exhibit notable advantages under comparable computational budgets. **We do not claim that 2D features can fully express 3D volumes;** rather, we make a more moderate point: at the current stage, through structured reassembly and voxel-level positional embeddings, a more generalizable 2D encoder can robustly carry 3D spatial information. This design does not "compress" dimensionality, but **preserves spatial structures and relations that remain interpretable from a 3D perspective, built on top of a 2D semantic backbone.** To avoid any misunderstanding that 2D encoders are categorically superior to 3D ones, we have added clarifications in **`Section 4.3 (ii)`**. We strictly present the empirical results without extrapolating claims beyond the available evidence. We sincerely appreciate the reviewer’s rigor.
>
> We also agree with the reviewer that "key slice + description" cannot serve as evidence that 2D information supplements 3D reasoning. Our intention was to highlight that slice-level interactions—localization, explanation, annotation, teaching—are continuous and unavoidable in medical LVLM applications. Maintaining continuity between 2D and 3D, rather than treating them as disjoint modules, better matches real-world workflow efficiency and usage patterns; empirically, this unification does not compromise 3D performance.
>
> We also strongly resonate with the reviewer’s question regarding whether true synergistic gains exist. To address this, we conducted 25% / 50% / 100% ablations under strictly balanced 2D/3D sample conditions. Mixed training indeed demonstrates synergistic benefits:
>
> |Training Strategy|Ratio|SLAKE|VQA-RAD|RadFig-VQA|M3D-VQA|CT-RATE|3D-RAD|
> |-|:-:|:-:|:-:|:-:|:-:|:-:|:-:|
> |2D-Only|25%|70.6|62.5|77.2|-|-|-|
> |3D-Only|25%|-|-|-|65.8|84.5|65.4|
> |Mixed|25%|72.2|62.5|78.2|69.9|84.9|66.9|
> |2D-Only|50%|75.8|66.7|79.4|-|-|-|
> |3D-Only|50%|-|-|-|72.7|86.2|67.2|
> |Mixed|50%|77.3|70.8|79.8|73.6|87.1|68.1|
> |2D-Only|100%|81.0|72.3|78.2|-|-|-|
> |3D-Only|100%|-|-|-|74.4|86.6|68.6|
> |Mixed|100%|81.2|71.8|81.9|74.7|86.6|69.3|
>
> Based on this result, we would like to explain that using a 2D encoder’s fine-grained semantic structuring to carry 3D features is still expressing 3D information. Under the currently attainable pretraining scale, 2D encoders exhibit more robust semantic generalization, enabling them to serve as a reliable semantic foundation for constructing 3D inputs while remaining naturally compatible with slice-level inputs. This allows the entire framework to maintain a coherent representational pathway within a unified architecture. We have incorporated this clarification in **`Section H.2 (iv)`**.
>
> Overall, we fully agree with the reviewer’s perspective and appreciate the feedback, which has helped us articulate the motivation and limitations of our method more clearly at this stage.
>
> [1] Li, W., Yuille, A., & Zhou, Z. (2025). How well do supervised 3d models transfer to medical imaging tasks?. arXiv preprint arXiv:2501.11253.

---

> ### Author Response · Authors · 2025-11-22
> **Author Response to Reviewer xwuZ - Round 2 (Part 2/2)**
>
> > Regarding W2
>
> We appreciate the reviewer’s follow-up question. In constructing the dataset, we strictly distinguished between the number of original images and the number of questions generated on top of them: the dataset contains 170,280 distinct 3D volumes (over 20M slices) and 327,063 distinct 2D slices, with no overlap in their sources—2D images are *not* obtained by slicing the 3D volumes.
>
> For 3D data, to fully utilize their spatial and semantic density, each volume is paired with an average of ~8 questions. In contrast, most 2D slices are associated with only a single question; only a few datasets (e.g., SLAKE, VQA-RAD) include multiple independent questions for the same image. We have **incorporated these clarifications** into the data section of the main paper.
>
> ---
>
> > Differences to fVLM
>
> We thank the reviewer for the constructive feedback. We understand the concern regarding potential similarities between fVLM and OSE at the mask-based aggregation level, and we agree that both involve region-mask–driven feature aggregation. However, the two differ substantially in their design goals and operational mechanisms. fVLM performs sample-level optimization via indiscriminate organ-level semantic contrastive learning, with the core objective of jointly optimizing the consistency between organ features and regional semantics. In contrast, OSE is task-aware: it uses the aggregated organ semantics as lightweight prompts that dynamically interact with global visual features, aiming to enhance the representation quality of key organs during LVLM training.
>
> Importantly, OSE’s adaptive aggregation strategy is complementary to global features. For large organs, we moderately compress their regions (e.g., for the lungs, retaining only ~20%–35% of the original spatial tokens) to obtain a more compact dual-scale representation. For small organs, we instead amplify their regions (e.g., kidneys expanded by ~8–10×) to prevent them from being overwhelmed by global features in the joint embedding space. We have validated the effectiveness of OSE through sensitivity analysis and ablations in **`Tables 11 and 12`** of the manuscript.
>
> We also appreciate the reviewer’s attention to experimental rigor. In our setup, we strictly controlled for a consistent training framework, identical 3D data conditions, and comparable token budgets, enabling a fair comparison across multiple 3D encoders; the results are reported in the **`Regarding W1`** section. Our intention is **not to argue that a 2D encoder is superior to native 3D encoders,** but rather to assess the feasibility—at the current stage—of constructing 3D semantics from structured 2D features extracted by strong pretrained encoders.
>
> ---
>
> > Q1) Dataset details
>
> We fully agree with the reviewer on the importance of dataset composition in establishing the research context. Following the suggestion, we have added the corresponding data sources and statistics to the revised version to ensure a clearer presentation of the benchmark background. These additions help better illustrate the structure of the dataset and provide stronger contextual support for subsequent analyses. To avoid any misalignment of figure references during the rebuttal phase, **we commit to visualizing the full dataset overview in the main paper after the rebuttal is completed.**
>
> ---
>
> > Q2) & Q3)
>
> We sincerely appreciate the reviewer’s positive feedback on our rebuttal—this is greatly encouraging for us!
>
> ---
>
> > Q4) Quantitative metric evaluation
>
> We appreciate the reviewer’s further feedback. We fully agree with the reviewer on the value of the RadBERT classifier, particularly for **quantitatively evaluating the abnormality subsets referenced in clinical reports.**
>
> In response to the reviewer’s request, we have added the evaluation metrics with the RadBERT classifier. Below are the comparative results between our method and several existing approaches:
>
> |**Model**|**Precision**|**Recall**|**F1**|
> |-|:-:|:-:|:-:|
> |RadFM|13.1|6.4|7.2|
> |M3D-LaMed-7B|8.1|2.5|3.5|
> |M3D-LaMed-4B|16.5|8.4|9.6|
> |CT-CHAT|24.3|38.8|27.2|
> |CT2Rep|41.6|38.1|36.7|
> |CT-AGRG|37.8|**55.4**|**42.1**|
> |OmniCT|**41.7**|36.5|36.3|
>
> OmniCT outperforms most volume-driven models and previous unified models, and it achieves performance comparable to models specifically designed for 3D CT report generation (e.g., CT2Rep and CT-AGRG). The corresponding comparison results have been added to the revised **`Section 4.1`**. We thank the reviewer for motivating this valuable experimental extension.
>
> ---
>
> We once again sincerely appreciate the reviewer’s high-quality feedback, which has provided us with important insights and constructive directions for improvement. We hope that the detailed responses above effectively address any remaining concerns.

---

### Official Review · Reviewer_RrqG · 2025-10-31

**Soundness:** 3
**Presentation:** 3
**Contribution:** 3
**Rating:** 6
**Confidence:** 3

**Summary:**

This paper introduces OmniCT, a unified Large Vision-Language Model (LVLM) for CT interpretation that jointly models 2D slice-based and 3D volume-based inputs within one framework.
The key motivation is that current medical LVLMs are fragmented:
- Slice-driven LVLMs (e.g., Med-VLM, LLaVA-Med) have good 2D generalization but lack spatial consistency across slices.
- Volume-driven LVLMs (e.g., CT-CHAT, M3D-LaMed) capture 3D structures but are coarse-grained and not compatible with slice inputs.

OmniCT aims to bridge this gap through three main innovations:
1. Spatial Consistency Enhancement (SCE):
Introduces Volumetric Slice Composition, Tri-axial Positional Encoding, and a Mixture-of-Experts (MoE) hybrid projection to unify 2D and 3D visual tokens. This allows both slices and volumes to share a common latent space while maintaining volumetric spatial consistency.
2. Organ-level Semantic Enhancement (OSE):
Injects anatomical priors via organ segmentation and region-of-interest masking, followed by adaptive feature aggregation to emphasize small organs and reduce redundancy in large ones.
3. MedEval-CT Benchmark:
A large-scale dataset (1.7M samples) and evaluation suite unifying 2D and 3D CT tasks with multi-level metrics and a standardized evaluation toolkit.

**Strengths:**

The paper is strongly motivated and addresses a real, clinically relevant gap in current medical LVLMs:
1. Strong and clinically grounded motivation and
2. Unified slice–volume modeling framework

It proposes a coherent architectural design (SCE + OSE + MoE projection) that enables shared representation learning between slice-driven and volume-driven inputs. the fragmentation between 2D slice-based and 3D volume-based CT understanding. Its unified slice–volume framework is conceptually coherent and technically well-designed, combining Spatial Consistency Enhancement (SCE), Organ-level Semantic Enhancement (OSE), and a Mixture-of-Experts hybrid projection to jointly model local detail and global volumetric context. The integration of tri-axial positional encoding and anatomy-guided ROI alignment contributes to more spatially consistent and semantically rich representations. Empirically, the work demonstrates high-quality experimentation and reporting, including detailed ablations, multi-task evaluations, and architecture analysis that isolate the effects of each proposed module. Finally, the paper is well-written and logically structured.

**Weaknesses:**

1. Conceptual novelty and differentiation:
The “unified slice-volume” design is valuable but not fundamentally new, similar goals have been pursued by Med-2E3, Med3DInsight, and hybrid 2D/3D LVLMs. Can the authors clarify more clearly articulate what OmniCT does differently (e.g., tri-axial positional encoding vs. cross-slice attention in Med-2E3, why tri-axial positional encoding is important? Is there any ablation for this?).

2. Architectural complexity vs. simplicity:
In the MoE routing, only 2 expert model is used for 2D and 3D accordingly. However, it is difficult for us whether the 3D expert is really learning the 3D representations from tokens, same as 2D. Can you clarify more on how to well distinguish these two experts are learning distinctive representation to each other?

**Questions:**

1. Methodological distinction:
How does the proposed SCE differ from Med-2E3’s dynamic cross-slice fusion or Med3DInsight’s encoder alignment? It will be great if there is an explicit mathematical or architectural comparison.
2. Organ-level supervision:
Purely curious. The OSE module depends on pre-computed segmentation masks (TotalSegmentor). Does this supervision introduce label bias? How does OmniCT perform if segmentation is noisy or missing? Is the aggregated token more than enough for fusion. Wondering if the performance will enhance while the aggregated tokens has a longer context length for concatenation?
3. Cross-modality transfer:
The paper claims OmniCT trained on 2D can generalize to 3D tasks. Is this because of the tri-axial positional encoding or MoE routing? It will be great to provide an ablation to separate their contributions.

---

> ### Author Response · Authors · 2025-11-20
> **Author Response to Reviewer RrqG (Part 1/3)**
>
> We thank the reviewer for the careful assessment and constructive comments. The reviewer acknowledged that our method is clinically strong motivated, structurally well-designed, and supported by high-quality experiments and reporting. Below, we provide detailed point-by-point responses to the questions raised.
>
> ---
>
> ### **(w1) Conceptual novelty and differentiation**
> We thank the reviewer for the insightful comments on our "unified slice–volume" design. Here we summarize two categories of existing unified LVLM approaches to provide a clear comparative perspective:
> - **Feature alignment and enhancement** (e.g., Med-2E3, Med3DInsight): these methods rely on dynamic slice fusion or feature distillation mechanisms, using 2D prior knowledge to guide 3D volume perception. Conceptually, they are not truly unified and remain essentially single-modality settings.
> - **Hybrid-input compatibility** (e.g., RadFM [1], OmniV-Med [2]): these approaches employ general shape adaptation or treat a volume as a video sequence, achieving engineering-level compatibility between slice and volume inputs; however, they do not establish an architecture with semantically consistent feature spaces.
>
> The fundamental difference between OmniCT and the aforementioned approaches is that **we establish geometrically–spatially–semantically aligned representations** from the very start of training.
>
> Specifically, our volumetric slice composition (VSC) and tri-axial positional embedding (TPE) enforce shared geometric encoding rules between slices and volumes, improving visual encoding efficiency and preserving spatial awareness across heterogeneous input modalities. The MoE hybrid projection (MHP) decouples slice–volume feature inputs at shallow layers to avoid conflicts, while leveraging shared parameters and the LLM backbone to align the semantic feature space; it also applies token-level unshuffle to regulate the token budget. In addition, we design an organ-level semantic enhancement (OSE) to trace back and strengthen token features that carry higher organ-level semantic loads. Together, these components ensure that unified perception is achieved both structurally and semantically.
>
> In addition, we would like to clarify that **cross-slice attention and TPE operate through entirely different mechanisms.** Cross-slice attention performs semantic enhancement from a 2D perspective, whereas TPE directly supplies voxel-level spatial references, enabling consistent perception across slice and volume inputs. The gains from TPE are already reported in Table 1 (the SCE column). To further validate the TPE mechanism, we include the following ablations:
>
> |**PE Strategy**|**Perf. 2D**|**Perf. 3D**|**Avg.**|
> |:-|:-:|:-:|:-:|
> |No PE|78.7|62.2|70.4|
> |Single-axial PE|78.2|62.8|70.5|
> |Tri-axial PE|80.1|66.2|73.2|
>
> The results confirm that the **performance gains stem from the injected spatial awareness** rather than from additional parameters.
>
> Taken together, OmniCT’s contribution is not a continuation of unified input handling or semantic post-hoc corrections, but a unified architecture that enforces consistency from geometry to semantics. This places it in clear conceptual and methodological contrast to existing approaches.
>
> [1] Wu, C., et al. "Towards generalist foundation model for radiology by leveraging web-scale 2d&3d medical data," in Nature Communications, vol. 16, no. 1, pp. 7866, 2025.
>
> [2] Jiang, S., et al. "Omniv-med: Scaling medical vision-language model for universal visual understanding," in arXiv preprint arXiv:2504.14692, 2025.

---

> ### Author Response · Authors · 2025-11-20
> **Author Response to Reviewer RrqG (Part 2/3)**
>
> ### **(w2) Architectural complexity vs. simplicity**
> We appreciate the reviewer’s attention to the expert separability in the MoE hybrid projection (MHP). We would like to clarify that MHP is not designed to explicitly learn high-level 2D or 3D representations, but rather to **provide the structural adaptation necessary for a unified semantic space,** ensuring that two types of visual features—differing in channel dimensionality and distribution—can be stably mapped into the LLM’s semantic space. In fact, the natural separability of the two experts is an inherent consequence of OmniCT’s heterogeneous-modality modeling and its optimization-path design:
> - **Feature construction differences:** Volumetric slice composition (VSC) and TPE require differentiated adaptation for heterogeneous slice–volume inputs. Before reaching the experts, the two modalities already diverge in their feature distributions and arrangement patterns.
> - **Decoupled optimization paths:** The 2D and 3D experts in MHP share only the final-layer parameters, while each updates its shallow-layer gradients independently based on its respective training samples.
>
> In the revised manuscript, we **`include a t-SNE visualization in Figure 6`** showing clear clustering between the two expert feature groups, further supporting their separability. Moreover, whether the 3D expert truly captures 3D representations can be indirectly assessed through task performance: in the 3D-RAD anomaly detection task—where spatial consistency and organ-level semantics are essential—OmniCT surpasses CT-CHAT and M3D-LaMed by 21.99 and 34.48 points, respectively (Table 3).
>
> These results indicate that the expert separability in MHP arises from inherent differences in input structure and optimization dynamics, rather than from any artificially introduced complexity.
>
> ---
>
> ### **(q1) Methodological distinction**
> We appreciate the reviewer’s attention to the methodological differences between SCE and Med-2E3 / Med3DInsight. To more clearly delineate these distinctions, we provide the following structural comparison:
>
> > Dynamic cross-slice fusion in Med-2E3
>
> Med-2E3 extracts features using separate 2D and 3D encoders, and employs the TG-IS module to score 2D slices based on text relevance, using these scores to enhance the 3D representation through weighted fusion:
>
> $$ F_{2D}=\mathcal{E}_{2D}(S), \quad F\_{2D}=\mathcal{E}\_{3D}(V),\quad\alpha=\text{TG-IS}(F\_{2D},T),\quad \tilde{F}\_{3D}=\text{Fusion}(\alpha,F\_{2D},F\_{3D})$$
>
> This strategy essentially uses 2D semantics to augment 3D features, rather than enabling the two input types to be represented equivalently within the same latent space.
>
> > Encoder alignment in Med3DInsight
>
> Med3DInsight projects 3D representations into the semantic space of a 2D LVLM through the PSAT module:
> $$F_{3D}=\mathcal{E}_{3D}(V),\quad\tilde{F}\_{3D}=\text{PSAT}(F\_{3D})\rightarrow \mathcal{H}\_{2D\text{-}LVLM},$$
> This approach belongs to distillation-based alignment, relying on the 2D semantic space to perform post-hoc correction on the 3D encoder outputs.
>
> In contrast, as clarified in **`(w1)`**, SCE establishes geometric–spatial–semantic alignment, directly providing the prerequisites for a shared latent semantic space:
> $$F\_{2D}=\mathcal{E}\_{2D}(S)\oplus \text{TPE}_{2D} ,\quad F\_{3D}=\mathcal{E}\_{2D}(\text{VSC}(V)) \oplus \text{TPE}\_{3D}, \quad \text{MHP}(F\_{2D},F\_{3D}) \rightarrow \mathcal{H}\_{LLM}$$
>
> Therefore, SCE and Med-2E3 / Med3DInsight differ substantially in their **problem setting, objectives, and architectural design.**

---

> ### Author Response · Authors · 2025-11-20
> **Author Response to Reviewer RrqG (Part 3/3)**
>
> ### **(q2) Organ-level supervision**
> We appreciate the reviewer’s questions regarding OSE. OSE treats organ segmentation as a structurally stable regional prior, rather than as a supervision signal. It is used to identify regions carrying higher semantic load for the current task, enabling the model to trace back more discriminative token groups while retaining global features. Our reliance is **on organ-level semantic consistency, not pixel-level boundary fitting;** under this setting, the high stability of TotalSegmentor (average Dice 94.3%) is sufficient to provide reliable region cues [3].
>
> Regarding potential label bias, Table 11 in the manuscript addresses the reviewer’s concern by evaluating worst-case scenarios—using random ROI region and removing ROI tokens. OmniCT remains stable under these conditions. We further add ablations across different compression ratios, showing that varying the number of aggregated tokens yields consistent gains within a certain range, followed by a degradation trend:
> |$\mathbf{m_{2D}}$|$\mathbf{m_{3D}}$|**Perf. 2D**|**Perf. 3D**|**Avg.**|
> |:-:|:-:|:-:|:-:|:-:|
> |0|0|78.7|62.2|70.4|
> |36|40|80.7|63.8|72.2|
> |81|90|81.5|66.2|73.8|
> |144|160|81.2|66.0|73.6|
> |225|250|80.6|65.5|73.1|
>
> Overall, OSE leverages a robust regional prior to deliver stable organ-level semantic enhancement. These results have been incorporated into **`H.2 (iv)`** of the revised manuscript.
>
> [3] Wasserthal, J., et al. "TotalSegmentator: robust segmentation of 104 anatomic structures in CT images," in Radiology: Artificial Intelligence, vol. 5, no. 5, pp. e230024, 2023.
>
> ---
>
> ### **(q3) Cross-modality transfer**
> We appreciate the reviewer’s question regarding the mechanism behind cross-modal generalization. We confirm that OmniCT’s cross-modal generalization is not driven by any single component; rather, it **emerges from the combination of the single-tower semantic space (SigLip backbone) and the MHP alignment mechanism.** The single-tower design ensures that slice and volume features are always embedded within the same semantic neighborhood, while MHP further learns how to project visual features into the LLM semantic space. This allows projection patterns learned from slices to transfer to volumetric representations, and vice versa.
>
> To disentangle their contributions, we compare the configurations of "dual-tower without MoE" and "single-tower with MoE" under identical training conditions:
> |**Training Strategy**|**Perf. 2D**|**Perf. 3D**|**Avg.**|
> |-|:-:|:-:|:-:|
> |SigLip + M3D-CLIP (w/o MHP)|34.6|30.6|32.6|
> |SigLip + Siglip (w/ MHP)|55.3|48.6|52.0|
>
> Therefore, OmniCT’s cross-modal generalization emerges from the synergy between the unified semantic space and the adaptive alignment mechanism. We have added the relevant clarification to **`Section 4.3 (i)`** for completeness. We also thank the reviewer for prompting this valuable extension.
>
> ---
>
> **`Summary:`** We once again sincerely thank the reviewer for the careful evaluation and constructive feedback. We have addressed each comment in detail and carried out thorough revisions based on these insights. We hope that the above responses alleviate the reviewer’s concerns and lead to a more positive assessment of the novelty and overall quality of our work.

---

> ### Author Response · Authors · 2025-11-26
> **Kind Check for Any Remaining Concerns**
>
> Dear Reviewer,
>
> I hope this message reaches you well. As the discussion period is moving into its final days, we wanted to follow up and ensure that all of your concerns have been thoroughly addressed. If there are any additional questions, clarifications, or suggestions you would like us to consider, please don’t hesitate to let us know.
>
> Your insights have been invaluable in improving our paper, and we remain fully committed to addressing any final points you may have.
>
> Thank you for your continued engagement and thoughtful feedback.
>
> Best regards,
>
> The Authors

---

### Meta-Review · Area_Chair_iBMJ · 2026-01-06

**Summary:**

The paper presents OmniCT, a unified Large Vision-Language Model (LVLM) framework designed for Computed Tomography (CT) analysis. OmniCT aims to bridge this gap through a unified modeling paradigm.

The proposed architecture introduces two key modules:

1. Spatial Consistency Enhancement (SCE): This module employs volumetric slice composition and tri-axial positional encodings to inject 3D spatial awareness into the model while maintaining compatibility with 2D slice inputs.

2. Organ-level Semantic Enhancement (OSE): This module leverages anatomical segmentation masks to perform adaptive feature aggregation.

Additionally, the paper makes a significant contribution to the data landscape by introducing MedEval-CT, a comprehensive dataset and benchmark suite. MedEval-CT comprises approximately 1.7 million slice-volume VQA samples across seven clinical task types, designed to ensure balance across organs and tasks. Extensive experiments demonstrate that OmniCT outperforms existing medical and general-purpose LVLMs on both 2D slice benchmarks (e.g., SLAKE, VQA-RAD) and 3D volume benchmarks (e.g., M3D, CT-RATE). The results suggest that the unified approach successfully combines the generalization capabilities of 2D models with the spatial reasoning required for 3D analysis.

Besides, the authors have provided detailed, point-by-point rebuttals that address most reviewer concerns, adding new experiments (e.g., RadBERT evaluation, cross-LLM backbone validation, robustness tests) and clarifications. While some philosophical disagreements about the problem framing (particularly from R2) may persist, the technical contributions, empirical results, and the utility of the new benchmark are clear and valuable. Acceptance is recommended contingent on the authors incorporating the promised clarifications and minor revisions from the rebuttal into the final manuscript.

**Reviewer Concerns:**

The primary concerns raised by reviewers focused on motivation, dataset transparency, and experimental baselines:

1.  **Motivation for 2D Integration:** Reviewer xwuZ questioned the fundamental need to integrate 2D slices, arguing that CT is inherently 3D and 3D-native models (like nnU-Net) are typically superior. The reviewer was skeptical about whether 2D data truly enhances 3D reasoning or if it is merely a workaround.
2.  **Dataset Details and Provenance:** Reviewers xwuZ and GKxu initially found the description of the MedEval-CT dataset (sources, licensing, overlap with testing data) insufficient in the main text. There were concerns about potential data leakage and the circularity of using Qwen-based models for both data filtering and the backbone.
3.  **Comparison of 2D vs. 3D Encoders:** Reviewers requested stronger comparisons against native 3D encoders (e.g., VideoMAE-3D) to substantiate the claim that the authors' 2D-based approach is empirically superior.
4.  **Novelty:** Reviewer RrqG questioned the conceptual novelty compared to prior fusion methods like Med-2E3.

**Reviewer Scores:**

The reviews reflect a generally positive reception, though with one skeptical voice regarding the core motivation.

*   **Reviewer RrqG:** Score 6 (Marginally above acceptance). Found the clinical motivation strong and the framework coherent. Appreciated the clarifications on novelty and the MoE mechanism during the rebuttal.
*   **Reviewer GKxu:** Score 6 (Marginally above acceptance). Praised the SCE/OSE design and the benchmarking effort. The reviewer’s concerns about dataset leakage and stronger 3D baselines were largely addressed by the comprehensive rebuttal.
*   **Reviewer xwuZ:** Score 4 (Marginally below acceptance). This reviewer remained the most critical, primarily regarding the philosophical motivation of 2D/3D unification. However, they engaged actively and acknowledged the value of the new RadBERT experiments and VQA anti-leakage tests, indicating they might raise their score (though I cannot see the final rating).

---

### Decision · Program_Chairs · 2026-01-26

Accept (Poster)